# Dynamics and regulatory roles of RNA m⁶A methylation in unbalanced genomes

**Shuai Zhang[1,2†], Ruixue Wang[1,2†], Kun Luo[1,2], Shipeng Gu[1,2], Xinyu Liu[1,2], Junhan Wang[1,2], Ludan Zhang[1,2], Lin Sun[1,2]***

[1]Key Laboratory of Cell Proliferation and Regulation Biology of Ministry of Education, College of Life Sciences, Beijing Normal University, Beijing, China; [2]Beijing Key Laboratory of Gene Resource and Molecular Development, College of Life Sciences, Beijing Normal University, Beijing, China

## eLife Assessment

This **valuable** study suggests that the dosage compensation complex and m6A act in a feedback loop in *Drosophila melanogaster*. The study provides integrated analyses of RNA sequencing and mapping data of the m6A RNA modification in the context of unbalanced genomes, which suggests that m6A modification status may influence H3K16Ac deposition through regulation of the acetyltransferase MOF. However, it is not clear whether this regulation is directly or indirectly related to m6A regulation. The evidence is considered **incomplete** due to technical concerns, as quantitative assessments were made using non-quantitative methods.

**\*For correspondence:**
sunlin@bnu.edu.cn

†These authors contributed equally to this work

**Competing interest:** The authors declare that no competing interests exist.

**Abstract** *N*⁶-methyladenosine (m⁶A) in eukaryotic RNA is an epigenetic modification that is critical for RNA metabolism, gene expression regulation, and the development of organisms. Aberrant expression of m⁶A components appears in a variety of human diseases. RNA m⁶A modification in *Drosophila* has proven to be involved in sex determination regulated by *Sxl* and may affect X chromosome expression through the MSL complex. The dosage-related effects under the condition of genomic imbalance (i.e. aneuploidy) are related to various epigenetic regulatory mechanisms. Here, we investigated the roles of RNA m⁶A modification in unbalanced genomes using aneuploid *Drosophila*. The results showed that the expression of m⁶A components changed significantly under genomic imbalance, and affected the abundance and genome-wide distribution of m⁶A, which may be related to the developmental abnormalities of aneuploids. The relationships between methylation status and classical dosage effect, dosage compensation, and inverse dosage effect were also studied. In addition, we demonstrated that RNA m⁶A methylation may affect dosage-dependent gene regulation through dosage-sensitive modifiers, alternative splicing, the MSL complex, and other processes. More interestingly, there seems to be a close relationship between MSL complex and RNA m⁶A modification. It is found that ectopically overexpressed MSL complex, especially the levels of H4K16Ac through MOF, could influence the expression levels of m⁶A modification and genomic imbalance may be involved in this interaction. We found that m⁶A could affect the levels of H4K16Ac through MOF, a component of the MSL complex, and that genomic imbalance may be involved in this interaction. Altogether, our work reveals the dynamic and regulatory role of RNA m⁶A modification in unbalanced genomes, and may shed new light on the mechanisms of aneuploidy-related developmental abnormalities and diseases.

## Introduction

Epigenetic modifications regulate gene expression in response to environmental changes and play important roles in the development of organisms and a variety of human diseases (*Jaenisch and Bird, 2003*; *Lence et al., 2017*). In addition to DNA and chromatin modifications, which are well studied, more than 100 RNA chemical modifications have been identified in cells to date (*Lee et al., 2014*; *Roundtree et al., 2017a*). As a marker of post-transcriptional regulation, RNA modifications participate in almost all aspects of RNA metabolism (*Roundtree et al., 2017a*). $N^6$-methyladenosine (m$^6$A) is the most prevalent internal modification in many eukaryotic messenger RNAs (mRNAs) and long noncoding RNAs (lncRNAs) (*Lee et al., 2014*; *Roundtree et al., 2017a*; *Yang et al., 2018*), which widely affects RNA alternative splicing (*Dominissini et al., 2012*), export (*Roundtree et al., 2017b*), stability (*Wang et al., 2014b*), and translation (*Meyer et al., 2015*). RNA m$^6$A modifications have been found to be enriched on the transcripts of genes that regulate development and cell fate specification (*Dominissini et al., 2012*; *Meyer et al., 2012*; *Geula et al., 2015*), and some m$^6$A sites are regulated in a tissue- or disease-specific manner (*Zaccara et al., 2019*). In addition, the abnormal expression of m$^6$A components is related to the tumorigenesis, proliferation, and metastasis of many types of cancers (*Pinello et al., 2018*; *Ma et al., 2019*). Therefore, it is of great significance to study RNA m$^6$A methylation for revealing the mechanisms of gene expression regulation and human diseases.

At present, there are few studies on RNA m$^6$A modification in *Drosophila*, possibly due to its relatively low abundance (m$^6$A/A<0.2%) (*Haussmann et al., 2016*), and the mutation of some m$^6$A component genes will affect their viability and fertility (*Hongay and Orr-Weaver, 2011*; *Haussmann et al., 2016*; *Kan et al., 2017*). However, as a model organism with specific genetic and developmental advantages, *Drosophila* remains an excellent tool for studying the roles of epigenetic modifications in gene regulation, individual development, and disease process. Several components of the *Drosophila* m$^6$A methyltransferase complex (*Ime4*, *dMettl14*, and *fl(2)d* constitute the core complex, with *vir* and *nito* acting as cofactors) and an m$^6$A reader protein (*Ythdc1*) have been identified, all of which have homologues in mammals (*Lence et al., 2017*). Deletion of the major methyltransferase gene *Ime4* or the reader *Ythdc1* causes locomotion defects, and the splicing of sex-determining factor *Sxl* is affected (*Haussmann et al., 2016*; *Lence et al., 2016*; *Kan et al., 2017*). However, homozygous mutations in *fl(2)d*, *vir*, and *nito* were lethal, suggesting that these subunits have important functions other than methylation (*Penn et al., 2008*; *Yan and Perrimon, 2015*; *Kan et al., 2017*; *Lence et al., 2017*). RNA m$^6$A modification has an obvious sexual dimorphism in *Drosophila*, and reduced m$^6$A levels severely decreased the survival of females. It is thought to be due to the derepression of *msl-2* caused by aberrantly spliced *Sxl*, which forms the male-specific lethal (MSL) complex that associates with the X chromosome (*Haussmann et al., 2016*).

Dosage compensation is a widespread phenomenon in unbalanced genomes (*Lucchesi, 2018*; *Birchler and Veitia, 2021*). The deletion or duplication of some chromosomes rather than the whole chromosome set leads to genomic imbalance, i.e., aneuploidy (*Orr et al., 2015*). Aneuploid variation is usually detrimental to organisms (*Birchler and Veitia, 2012*; *Orr et al., 2015*; *Birchler and Veitia, 2021*), and is associated with developmental abnormalities, mental retardation, and various congenital defects (*Williams et al., 2008*; *Huang et al., 2021*; *Sanchez-Pavon et al., 2021*), possibly due to disorders in their gene expression systems (*Prestel et al., 2010*; *Letourneau et al., 2014*). Studies across species have pointed out that there is genome-wide *trans* modulation in aneuploidy, and the genes on the varied chromosomes are compensated to a certain extent, while genes located on the rest of the genome are mainly regulated in the opposite direction to the changes of chromosome numbers, which is known as the inverse dosage effect (*Birchler and Veitia, 2012*; *Birchler and Veitia, 2021*; *Sun et al., 2013c*; *Hou et al., 2018*; *Shi et al., 2021*). Histone modification (*Zhang et al., 2021a*), chromatin remodeling (*Birchler, 2016*), lncRNAs (*Zhang et al., 2023*), and microRNAs *Shi et al., 2022* have all been shown to play a role in genomic imbalance.

Because of the haploinsufficiency for X-linked genes, heterogametic individuals in organisms with XY sex determination systems could be regarded as analogous to aneuploidy (*Disteche, 2016*), including humans and *Drosophila*. Some studies have linked histone H4 lysine 16 acetylation (H4K16Ac) and non-coding *roX* RNAs to the dosage compensation of *Drosophila* (*Park et al., 2010*; *Conrad et al., 2012*). On the other hand, the compensation of X chromosome in human is thought to be regulated by lncRNA X-inactive specific transcript (*XIST*) (*Jordan et al., 2019*). Interestingly, in *Drosophila*, RNA m$^6$A modification indirectly affects gene expression through *Sxl* and *msl-2*; while in

human, m⁶A methylation of the key lncRNA *XIST* is necessary for the silencing of gene transcription on one of the female X chromosomes (*Patil et al., 2016*). Moreover, most tumor cells have genomic instability and high levels of aneuploidy (*Ben-David and Amon, 2020*; *Chiarle, 2021*), and at the same time, m⁶A component genes are often aberrantly expressed in various cancers (*Pinello et al., 2018*; *Ma et al., 2019*).

An increasing number of studies have found that dosage-related effects in aneuploidy may be the integration of multiple modulations rather than through a single mechanism (*Prestel et al., 2010*; *Birchler, 2016*). The effects of genomic imbalance are complicated, and the model of dosage compensation and global gene regulation in *Drosophila* has been extended to autosomal aneuploidies and sex chromosome aneuploid metafemales where MSL complexes are not assembled (*Sun et al., 2013b*; *Sun et al., 2013c*; *Zhang et al., 2021b*). In addition, genomic imbalance and *trans* regulatory mechanisms in other species such as maize, *Arabidopsis*, and humans have also been investigated (*Hou et al., 2018*; *Raznahan et al., 2018*; *Shi et al., 2021*; *Yang et al., 2021*; *San Roman et al., 2023*). To reveal the role of RNA m⁶A modification in unbalanced genomes, we studied the dynamic changes and regulatory functions of m⁶A methylation under genetic imbalance conditions using autosomal and sex chromosome aneuploid *Drosophila* maintained in our laboratory. Meanwhile, dosage-sensitive modifiers, differential alternative splicing events, the MSL complex, and other factors that may mediate the relationships between RNA m⁶A modification and the dosage-related effects of aneuploidy were also investigated. In summary, we provided a comprehensive picture of RNA m⁶A methylation in unbalanced genomes.

## Results

### The responses of m⁶A components under genomic imbalance

RNA m⁶A methylation is a reversible epigenetic modification, and its dynamic process is mediated by m⁶A methyltransferases (writers), demethylases (erasers), and m⁶A recognition proteins (readers) (*Lee et al., 2014*; *Yang et al., 2018*; *Zaccara et al., 2019*; *Figure 1A*). We first detected the expression of m⁶A components in *Drosophila* larvae with karyotypes of normal diploids and trisomies that have an additional chromosome arm using RT-qPCR to determine whether RNA m⁶A dynamics are affected by correlative enzymes in unbalanced genomes (*Figure 1B*; *Figure 1—figure supplement 1A and B*). It was found that the transcription levels of most m⁶A writers and the major m⁶A reader are downregulated in aneuploids compared with their respective sex-corresponding controls (*Figure 1B*). Unlike the other components, the cofactor *vir* of methyltransferase complex is up-regulated in trisomy 2L females (*Haussmann et al., 2016*; *Lence et al., 2016*; *Kan et al., 2017*). Because RNA m⁶A modification is enriched in the nervous system of *Drosophila*, the brains of aneuploid third instar larvae were also used to detect the expression of m⁶A components. As expected, a decreased expression trend of m⁶A components similar to that of the whole larvae was observed (*Figure 1—figure supplement 1C*).

Previous studies have pointed out that the m⁶A methylomes have temporal and spatial specificity in the development process of organisms, especially for embryonic development and cell differentiation (*Meyer and Jaffrey, 2014*; *Wang et al., 2014a*). Aberrant expression of m⁶A components may cause defects in embryogenesis and even early embryonic lethality (*Zhong et al., 2008*; *Wang et al., 2014a*; *Geula et al., 2015*; *Zhao et al., 2017*). Therefore, we designed probes of m⁶A components (*Figure 1—figure supplement 1D and E*) to examine the mRNA expression and localization patterns during early embryonic development of aneuploid *Drosophila* using high-resolution tyramide signal amplification-based fluorescence in situ hybridization (TSA-FISH) (*Lécuyer et al., 2007*; *Jandura et al., 2017*). The results showed that the subembryonic distribution patterns of the five components of m⁶A methyltransferase complex and one m⁶A reading protein are similar in wildtype and aneuploidies (*Figure 1C and D*; *Figure 1—figure supplement 2A–F*). At the blastoderm stage (stage 1–5), the probe signals of m⁶A components are widely distributed in the surface cell layer, yolk plasma, yolk cortex, and show a pattern of basal enrichment (*Figure 1—figure supplement 2A–F*). For the gastrulae at stage 6–11, the transcripts of m⁶A components are mainly located in head, amnioproctodeal invagination, germ band, and as in previous studies (*Lence et al., 2016*), and show an enrichment in neuroectoderm (*Figure 1C*; *Figure 1—figure supplement 2A–F*). At later stages of embryonic development (stages 12–13 and 14–17), the probes are distributed in brain, ventral nerve cord, midgut, and salivary gland (*Figure 1C*; *Figure 1—figure supplement 2A–F*).

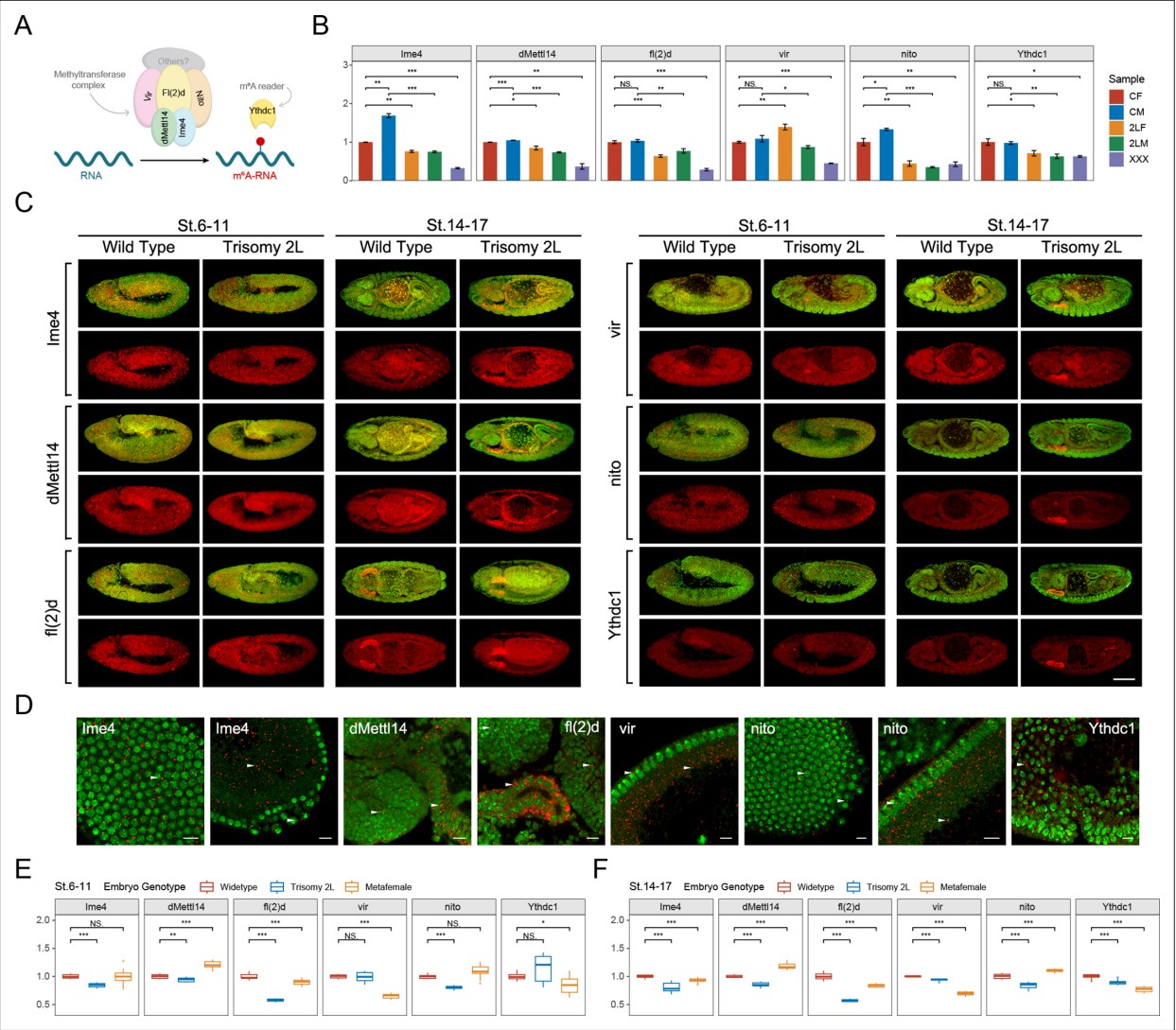

**Figure 1.** The responses of m⁶A methyltransferases and reader protein under the condition of genomic imbalance. (**A**) Schematic diagram of m⁶A components in *Drosophila*. (**B**) RT-qPCR analysis of messenger RNA (mRNA) levels of m⁶A methyltransferases and reader protein in third instar larvae of wildtype and trisomy *Drosophila*. CF, wildtype female control; CM, wildtype male control; 2LF, trisomy 2L female; 2LM, trisomy 2L male; XXX, metafemale; 2L, chromosome 2 left arm. Sample size = 3. Student's *t* test *p<0.05, **p<0.01, ***p<0.001. (**C**) Subembryonic distribution patterns of the transcripts of m⁶A components in wildtype and trisomy 2L *Drosophila*. The names of the genes were shown in the left of the pictures; the genotypes and stages were shown above. Red, probes; green, DAPI. Scale bar, 100 μm. (**D**) Subcellular localization of probe signals. Probe name was written in the corner of each picture. Red, probe; green, DAPI. Arrowheads indicate the foci of probe signals. The tissue types are (1) blastoderm nuclei; (2) yolk plasm and pole cells; (3) brain and midgut; (4) salivary gland and midgut; (5) blastoderm nuclei and yolk cortex; (6) blastoderm nuclei and pole cells; (7) blastoderm nuclei and yolk cortex; (8) germ band. Scale bars, 10 μm. (**E,F**) The expression levels of m⁶A component genes in stage 6–11 (**E**) and stage 14–17 (**F**) represented by relative fluorescence intensity of probes compared with DAPI signals. The expression of wildtype embryos was set as one. Sample size = 10. Student's *t* test *p<0.05, **p<0.01, ***p<0.001.

The online version of this article includes the following figure supplement(s) for figure 1:

**Figure supplement 1.** Primers of m⁶A methyltransferases and reader protein for RT-qPCR and TSA-FISH.

**Figure supplement 2.** Embryo tyramide signal amplification-based fluorescence in situ hybridization (TSA-FISH) of m⁶A components in wildtype, trisomy, and MSL2-overexpressed trisomy *Drosophila*.

By observing the subcellular localization of the probe signals of m⁶A components at higher magnification, it can be found that most of the mRNAs of these genes have nuclei-associated localization patterns (*Figure 1D*; *Figure 1—figure supplement 2A′–F′*). In the early and late stages of embryonic development, the probe signals of *Ime4*, *dMettl14*, *fl(2)d*, *vir*, and *Ythdc1* form dense small foci near the nucleus, which is a perinuclear distribution (*Figure 1D*; *Figure 1—figure supplement 2A′–F′*). For

*nito*, in addition to the perinuclear signals, there is also an obvious signal of intranuclear localization during early embryogenesis, which is manifested as one or two small foci in the blastoderm nucleus (*Figure 1D*; *Figure 1—figure supplement 2E'*).

Although there seems to be no difference in the localization of the transcripts of m$^6$A components that we detected in aneuploidy and wildtype embryos, the expression levels of these genes, as determined by relative fluorescence intensity, are significantly changed (*Figure 1E and F*; *Figure 1—figure supplement 2A''–F''*). Except for the irregular fluctuations at early embryonic stages, the expression levels of most m$^6$A components in aneuploids are lower than those in wildtype at more mature stages (*Figure 1E and F*; *Figure 1—figure supplement 2A''–F''*), which is similar to the trend we detected in third instar larvae (*Figure 1B*; *Figure 1—figure supplement 1C*). The transcripts of m$^6$A methyltransferase and reader protein have specific subembryonic and subcellular localization patterns in the development of *Drosophila* embryos, which may be a mechanism to regulate cellular functions, and ensure appropriate cell growth and differentiation (*Lécuyer et al., 2007*).

## Genome-wide mapping of RNA m$^6$A methylation in aneuploid *Drosophila*

Subsequently, we detected the global levels of RNA m$^6$A methylation in wildtype and aneuploid *Drosophila* larvae to determine whether m$^6$A abundance is altered under the condition of genomic imbalance due to modulation by transmethylases. The overall abundance of m$^6$A was represented by the m$^6$A/A ratio in total RNA (*Figure 2A*). It was found that the m$^6$A abundance of wildtype males is higher than that of females, but both are lower than 0.2%, which is consistent with the finding that m$^6$A levels in *Drosophila* are relatively low (*Haussmann et al., 2016*; *Lence et al., 2016*). Females with triple chromosome 2 left arms (2L) and metafemales with triple X chromosomes have significantly higher m$^6$A abundances than diploid females, whereas there is no significant difference between trisomy 2L males and wildtype males (*Figure 2A*). Therefore, genomic imbalance can affect the m$^6$A methylation status to some extent, and this epigenetic modification is different between males and females. However, the m$^6$A abundance in aneuploidies did not follow the expression levels of transmethylases.

To obtain m$^6$A mapping across the transcriptome of aneuploid *Drosophila*, whole larvae with different karyotypes were used for m$^6$A methylated RNA immunoprecipitation sequencing (MeRIP-Seq). By identifying consistent peaks in two biological replicates, approximately 10,000 m$^6$A peaks were found in wildtype females and males, whereas there were less than half the number of methylation sites in the three kinds of aneuploidies (*Figure 2B*). When these m$^6$A sites were annotated to the genes, the changes in the number of m$^6$A-marked genes in samples of different genotypes were in accordance with that of m$^6$A peaks, in which there are about 2500 m$^6$A-marked genes in wildtype and about 1000 m$^6$A-modified genes in trisomies (*Figure 2C*). Both the number of m$^6$A peaks and the number of m$^6$A-modified genes are the least in trisomy 2L females (*Figure 2B and C*). The changes in the number of RNA methylation sites in aneuploids did not coincide with the changes of overall m$^6$A abundance, but instead matched the expression of m$^6$A components. We speculate that this may be caused by the nonuniformity and heterogeneity of RNA m$^6$A modification, including the tissue specificity, the developmental specificity, the different numbers of m$^6$A sites in one transcript, the different proportions of methylated transcripts, etc. (*Meyer et al., 2012*; *Meyer and Jaffrey, 2014*; *Zaccara et al., 2019*). Counting the number of reads that mapped to m$^6$A peaks in wildtype and aneuploidy MeRIP samples showed that the read levels at m$^6$A sites in aneuploidies are significantly higher than that in wildtypes (Mann-Whitney U test p-values<0.001; *Figure 2D*); further analysis of the relative IP/input ratios showed that the values of aneuploidies are still higher than wildtypes (Mann-Whitney U test p-values<0.001), which confirmed our hypothesis. These results suggest that RNA m$^6$A modification exhibits greater heterogeneity in unbalanced genomes.

We next studied the overall characteristics of m$^6$A methylation in aneuploid and control *Drosophila*. m$^6$A peaks are localized along all autosomes and sex chromosomes, and the numbers of peaks are correlated with chromosome length (*Figure 2—figure supplement 1A*). Methylation sites are widely distributed on 5'UTR (30–50%), 3'UTR (10–15%), promoter (15–20%), and internal exons (20–35%), and the aneuploidies appeared to have a higher proportion of 5'UTR peaks and a lower proportion of exon peaks than the wildtype (*Figure 2E*). The highest values of density distributions of the length of exons containing m$^6$A peaks are around 250 bp, and approximately 90% of the exons are longer

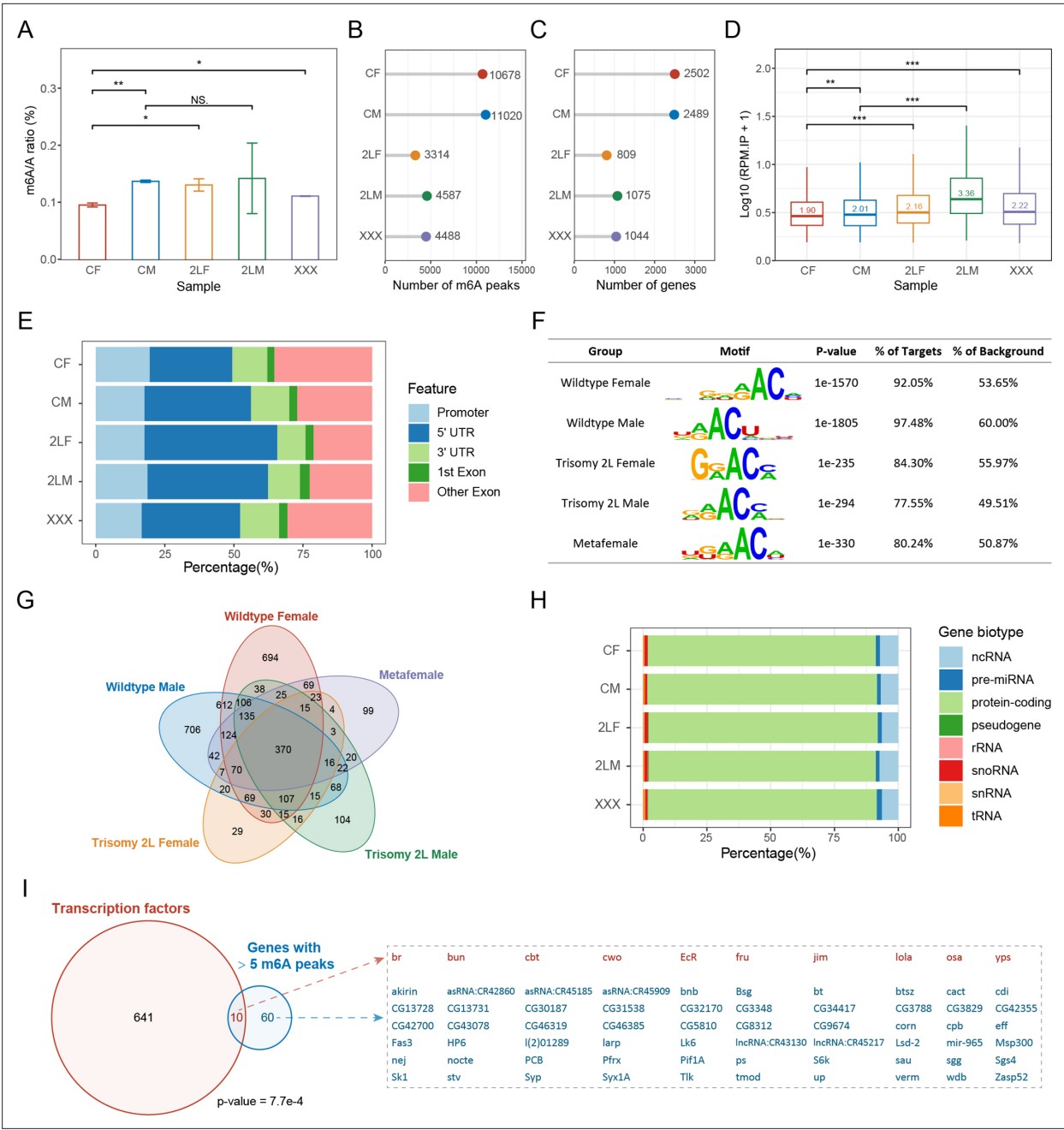

**Figure 2.** Overview of RNA m⁶A methylation in aneuploid *Drosophila*. (**A**) Global m⁶A abundance in third instar larvae of wildtype and aneuploid *Drosophila*. Data represent the mean of two independent experiments, each containing three or four biological replicates. Quantification was performed using EpiQuik m⁶A RNA Methylation Quantification Kit. Error bar indicates the standard error of the means (SEM). Student's *t* test *p<0.05, **p<0.01. (**B**) The number of m⁶A peaks identified by MeRIP-Seq. (**C**) The number of m⁶A-modified genes obtained by annotating the peaks. (**D**) Expression levels of m⁶A modification sites in IP samples expressed as log10-transformed reads per million (RPM). The number on the boxplot indicates the median RPM of each sample. Mann-Whitney U test *p<0.05, **p<0.01, ***p<0.001. (**E**) Percentages of peaks localized on different gene features. (**F**) The most enriched motifs obtained by de novo motif analysis of the m⁶A peaks. (**G**) Venn diagram showing the intersection of m⁶A-modified genes in each sample. (**H**) Gene biotypes of m⁶A-modified genes. (**I**) Venn diagram showing the intersection of transcription factors and m⁶A-modified genes with more than five m⁶A peaks in all samples. Seventy genes with more than five m⁶A peaks were listed on the right, with transcription factors in red and others in blue. p-Value indicates one-tailed Fisher's exact test. CF, wildtype female control; CM, wildtype male control; 2LF, trisomy 2L female; 2LM, trisomy 2L male; XXX, metafemale; 5'UTR, 5' untranslated region; 3'UTR, 3' untranslated region; MeRIP-Seq, m⁶A methylated RNA immunoprecipitation sequencing.

The online version of this article includes the following figure supplement(s) for figure 2:

*Figure 2 continued on next page*

*Figure 2 continued*

**Figure supplement 1.** Characteristics of m⁶A-modified genes in wildtype and aneuploid *Drosophila*.

**Figure supplement 2.** Functional and pathway analysis of m⁶A-modified genes in wildtype and aneuploid *Drosophila*.

than the typical 140 bp (*Figure 2—figure supplement 1B*). De novo motif analysis of m⁶A methylation peaks in each sample showed that the top-ranked motif is consistent with the conserved m⁶A motif DRACH (D=G/A/U, R=G/A, H=U/A/C; *Figure 2F*). In addition, enrichment analysis of the known motif DRACH showed that it is significantly enriched in all samples, and more than 97% of the m⁶A sites have this sequence (*Figure 2—figure supplement 1C*).

There are 370 common genes to which the m⁶A peaks were annotated in all genotypes (*Figure 2G*). We classified the m⁶A-modified genes in each sample and found that they were mostly protein-coding genes, whereas only about 7% were ncRNAs (*Figure 2H*). As described in previous studies (*Meyer et al., 2012*; *Zaccara et al., 2019*), most methylated RNAs have one m⁶A peak (*Figure 2—figure supplement 1D–H*). However, there are still some genes whose transcripts can be highly methylated and contain more than five m⁶A sites (*Figure 2—figure supplement 1D–I*). It was found that transcription factors were significantly enriched in 70 genes with more than five m⁶A peaks in both wildtype and aneuploidy (Fisher's exact test p-value = 7.7e-4; *Figure 2I*; *Figure 2—figure supplement 1I*). Thus, specific types of genes may be preferentially targeted by m⁶A modification, such as transcriptional regulators (*Kan et al., 2017*). Analysis of the functions of m⁶A-modified genes in each sample revealed that they were enriched for a large number of common functions, among which the most significant were those related to morphogenesis, development, and growth (*Figure 2—figure supplement 2A*). The 141 GO terms shared by the five genotypes were summarized and found to include functions about metabolism, regulation, cellular processes, signaling pathways, behavior, and immune response (*Figure 2—figure supplement 2B and C*). Through pathway enrichment analysis of m⁶A-modified genes, nine terms were found to be consistently enriched in all samples, including MAPK signaling pathway, Hippo signaling pathway, TGF-β pathway, etc. (*Figure 2—figure supplement 2D–F*).

## DMPs and their associated genes

We then searched for the differentially methylated peaks (DMPs) in autosomal and sex chromosome aneuploidies compared with wildtype *Drosophila* of their corresponding sex. The number of DMPs and the changing trend of DMP methylation status are shown in the figures (*Figure 3A–D*). In trisomy 2L females, the number of up-regulated DMPs (1131) is less than that of down-regulated DMPs (1602). In trisomy 2L males, the number of up-regulated DMPs (1521) is slightly higher than that of down-regulated DMPs (1259). In metafemales, the number of up-regulated DMPs (1393) is higher than that of down-regulated DMPs (926) (*Figure 3A–D*). More DMP-associated genes were found in autosomal aneuploidies than in sex chromosome aneuploidy (trisomy 2L female = 2148, trisomy 2L male = 2055, metafemale = 1771; *Figure 3E*). Also, the total number of peaks after the differential analysis would be different from the number of m⁶A peaks identified in the previous section, because the analysis software merged and recalculated the peaks (*Stark and Brown, 2013*).

DMPs are distributed on all chromosomes (*Figure 3—figure supplement 1A*). Specifically, the MeRIP-Seq signal intensity at the location of DMPs showed that the m⁶A-marked reads in trisomies are denser than those of wildtypes, especially on the triple chromosomes (*Figure 3F–H*). The density of reads on chromosome 2L of trisomy 2L *Drosophila* is higher than that on other chromosomes, while the density of reads on chromosome X of metafemales is significantly higher than that of autosomes (*Figure 3F–H*). Previous studies on aneuploid *Drosophila* showed that the expression levels of genes on chromosomes with increased copy numbers were mostly similar to those in diploids, and the expression of genes on unvaried chromosomes were widely inverse regulated (*Sun et al., 2013c*; *Birchler, 2016*; *Zhang et al., 2023*). It can be concluded that the changes in the density of reads at DMP sites are not due to a direct gene dosage effect. Similar results were obtained by counting the methylation status of cis and trans transcripts (*Figure 3I–K*). Overall, the transcripts of genes located on the varied chromosomes in three trisomies have more up-regulated DMPs than down-regulated DMPs, which is more obvious in metafemales (*Figure 3I–K*). In trisomy 2L females and males, genes

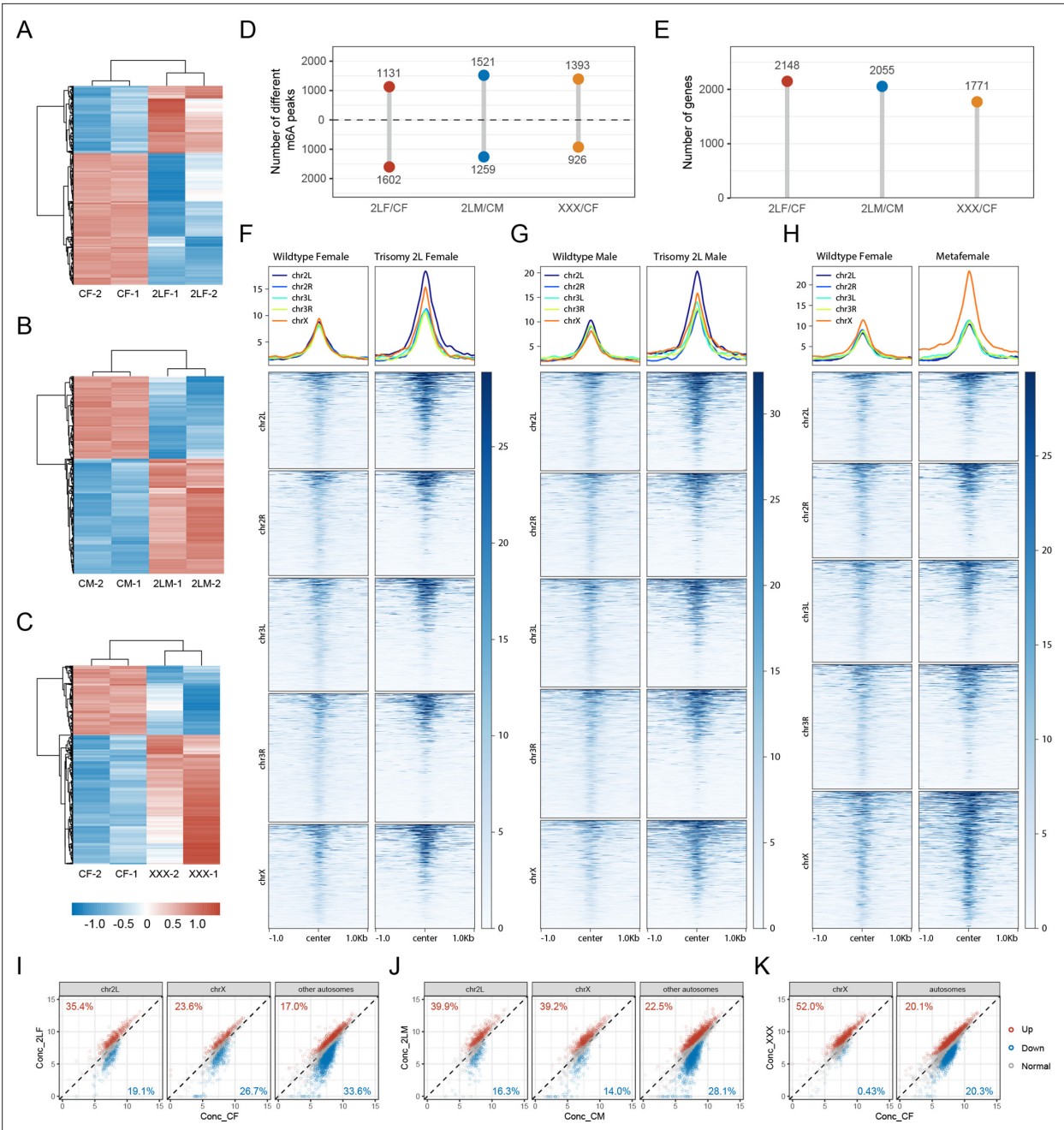

**Figure 3.** Differential m⁶A methylome analysis of aneuploid *Drosophila*. (**A–C**) Heatmaps of differentially methylated peaks (DMPs) in trisomy 2L females (**A**), trisomy 2L males (**B**), and metafemales (**C**), and their corresponding control groups. The threshold of significance was p-value≤0.1. (**D**) The number of DMPs. The threshold of significance was set to p-value≤0.1. The numbers above the horizontal dashed lines indicate peaks with up-regulated methylation levels, and the numbers below indicate peaks with down-regulated methylation levels. (**E**) The number of DMP-associated genes. (**F–H**) Profiles and heatmaps illustrating the density of m⁶A-modified reads at the DMP positions in trisomy 2L females (**F**), trisomy 2L males (**G**), metafemales (**H**), and their corresponding controls. The DMPs were divided into five groups according to the chromosomes they located. (**I–K**) Scatter plots showing the concentration of reads at methylation sites on different chromosomes in trisomy 2L females (**I**), trisomy 2L males (**J**), and metafemales (**K**). Red points indicate significantly up-regulated m⁶A peaks, blue points indicate significantly down-regulated m⁶A peaks, and gray points indicate m⁶A peaks without significant changes. The percentages of DMPs on *cis* and *trans* chromosomes were indicated in the corners of the plots. CF, wildtype female control; CM, wildtype male control; 2LF, trisomy 2L female; 2LM, trisomy 2L male; XXX, metafemale.

The online version of this article includes the following figure supplement(s) for figure 3:

**Figure supplement 1.** Characteristics of differentially methylated peak (DMP) associated genes.

localized on other autosomes have a higher proportion of down-regulated DMPs, while the X-linked *trans* genes show sexual dimorphism and X chromosome-specific response to genomic imbalance (*Figure 3I and J*). For metafemales, the transcripts of *trans* genes have similar numbers of up-regulated and down-regulated DMPs (*Figure 3K*). Therefore, *cis* genes in trisomy generally possessed a higher proportion of up-regulated DMP-associated transcripts, whereas the methylation states of *trans* genes are variable, depending on the identities of varied chromosomes and their genomic locations. The above results give us reason to speculate that the changes of m⁶A methylation may affect dosage compensation of *cis* genes and inverse dosage modulation of *trans* genes in aneuploidy.

The distribution of DMPs along gene features includes 5'UTR (30–35%), 3'UTR (20–25%), promoter (10–25%), and internal exon (25–35%), among which the proportion of 3'UTR is higher than that of all m⁶A sites (*Figure 3—figure supplement 1B*). The length distributions of exons with DMPs are similar to that of all m⁶A-modified exons (*Figure 3—figure supplement 1C*). De novo motif analysis of the sequences where DMPs are located or enrichment analysis of the known motif DRACH both showed that DMP sites are enriched for the conserved motif of m⁶A (*Figure 3—figure supplement 1D and E*). There are also similarities between the types of DMPs-associated genes and m⁶A-marked genes (*Figure 3—figure supplement 1F*). The DMP-associated genes of the three aneuploidies have complex overlapping relationships (*Figure 3—figure supplement 1G*). Most DMP-associated genes have one differentially methylated site, and only a few genes contained more than three DMPs (*Figure 3—figure supplement 1H–J*). There are 1042 common DMP-associated genes in all trisomies, of which 71 were transcription factors and were significantly enriched (Fisher's exact test p-value = 6.9e-5; *Figure 4A and B*). We speculate that these differentially methylated transcription factors may play an important role in the regulation of gene expression and development of aneuploidy.

Functional enrichment analysis showed that genes with up-regulated DMPs in three trisomies were enriched for 153 common functions, while the genes with down-regulated DMPs shared only 6 functions (*Figure 4—figure supplement 1A and B*). Similarly, genes with up-regulated DMPs were enriched for more consistent pathways in all aneuploids (*Figure 4—figure supplement 1C and D*). Most of the functions of DMP-associated genes are the same as those of m⁶A-marked genes, and these genes are also enriched for cell fate commitment, dorsal/ventral pattern formation, sex differentiation, post-transcriptional regulation of gene expression, and other additional GO terms. We found that many DMP-associated genes have functions related to gene expression regulation and dosage compensation, including some components of MSL complex (*msl-1*, *msl-3*, *lncRNA:roX1*) and the major sex-determining factor *Sxl* (*Figure 4C*). We also noticed that m⁶A component genes *fl(2)d*, *nito*, and *Ythdc1* themselves are differentially methylated in aneuploidy (*Figure 4C*), e.g., there was a common significantly up-regulated m⁶A peak in the 5'UTR of the transcripts of *fl(2)d* in three aneuploidies (*Figure 4D*). These data suggest that m⁶A dynamics in aneuploids may be involved in the expression regulation of unbalanced genomes through various genes with regulatory functions.

## Relationships between m⁶A and dosage-related modulation of gene expression in aneuploidy

To explore the relationships between RNA m⁶A modification and gene expression regulation under genomic imbalance, we combined the results of MeRIP-Seq and RNA-seq, and performed a comprehensive analysis. The results showed that about 12–15% of genes differentially expressed in aneuploid *Drosophila* also belong to DMPs-associated genes (*Figure 5A–C*). However, except for trisomy 2L males, the DEGs of the other two trisomies did not show enrichment of differential m⁶A methylation (Fisher's exact test p-values: trisomy 2L male = 0.012, trisomy 2L female and metafemale >0.05). There are 55 genes that are both differentially expressed and differentially methylated in all aneuploidies (*Figure 5D*). By analyzing the functions of genes that are simultaneously differentially expressed and differentially methylated, we obtained a number of consistent biological functions, including post-embryonic animal morphogenesis, cell communication, signal transduction, response to stimulation, and so on (*Figure 5E*), possibly reflecting the important roles of m⁶A-regulated expression in cell signaling and development of organisms.

By analyzing the crossover of different groups of m⁶A-modified genes and genes with canonical dosage effect, dosage compensation, and inverse dosage effect, we further revealed the relationships between m⁶A-modified genes and dosage-related effects in aneuploidy (*Figure 5F–H*). The results showed that in trisomy 2L females, *cis* dosage compensation genes, *trans* autosomal dosage

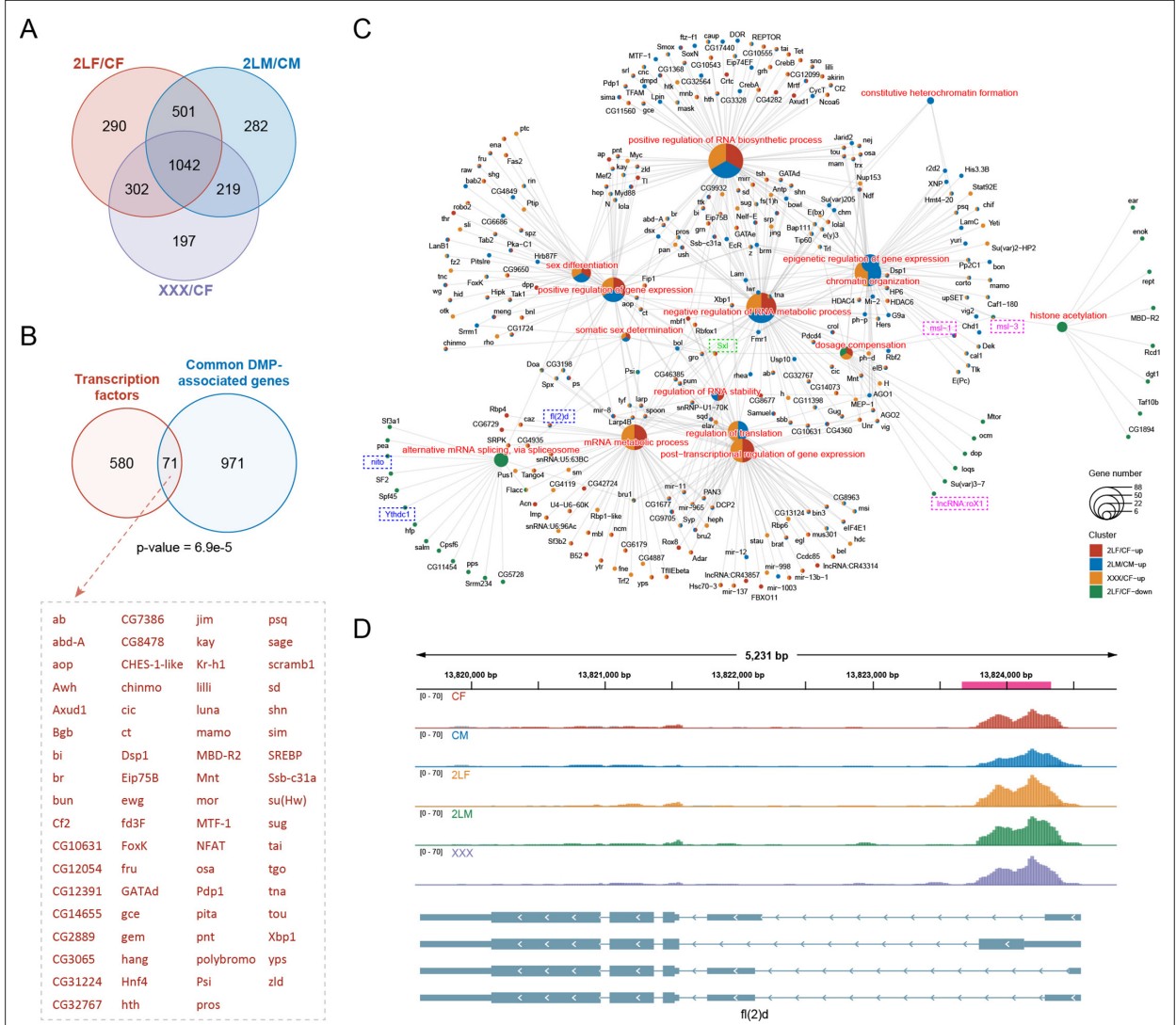

**Figure 4.** Differentially methylated peak (DMP) associated genes and their functions. (**A**) Venn diagram showing the number of common DMP-associated genes in three types of aneuploidies compared with wildtypes. (**B**) Venn diagram showing the intersection of transcription factors and the common DMP-associated genes in all comparisons. Transcription factors with DMPs were listed on the right. p-Value indicates one-tailed Fisher's exact test. (**C**) Network showing the functions related to expression regulation and dosage compensation enriched by DMP-associated genes. The color of the nodes indicates the comparison, and the size of the function nodes represents the number of DMP-associated genes connected with them. (**D**) Genome browser example of *fl(2)d* for indicated m⁶A methylated RNA immunoprecipitation sequencing (MeRIP-seq) data. Steelblue color represents input reads, while other colors represent IP reads. Signals were displayed as the mean counts per million (CPM) of two biological replicates. The gene architectures were shown at the bottom. The magenta rectangles at above represent DMP. CF, wildtype female control; CM, wildtype male control; 2LF, trisomy 2L female; 2LM, trisomy 2L male; XXX, metafemale.

The online version of this article includes the following figure supplement(s) for figure 4:

**Figure supplement 1.** Functional and pathway analysis of differentially methylated peak (DMP) associated genes.

effect genes, and *trans* unchanged genes are significantly enriched in m⁶A group with methylation in both trisomy and control; meanwhile, *cis* dosage effect genes, other autosomal dosage effect genes, and X-linked dosage effect genes are enriched in the group without methylation in both trisomy and control (***Figure 5F***). On the contrary, in trisomy 2L males, *cis* dosage effect genes, *trans* dosage effect genes, and *trans* unchanged genes are significantly enriched in m⁶A group whose genes are methylated in both trisomy and control; while the group of genes that are not methylated at all is enriched for *cis* dosage effect genes, *cis* dosage compensation genes, *trans* inverse dosage effect genes, and other autosomal dosage effect genes (***Figure 5G***). Metafemales performed similarly to trisomy 2L

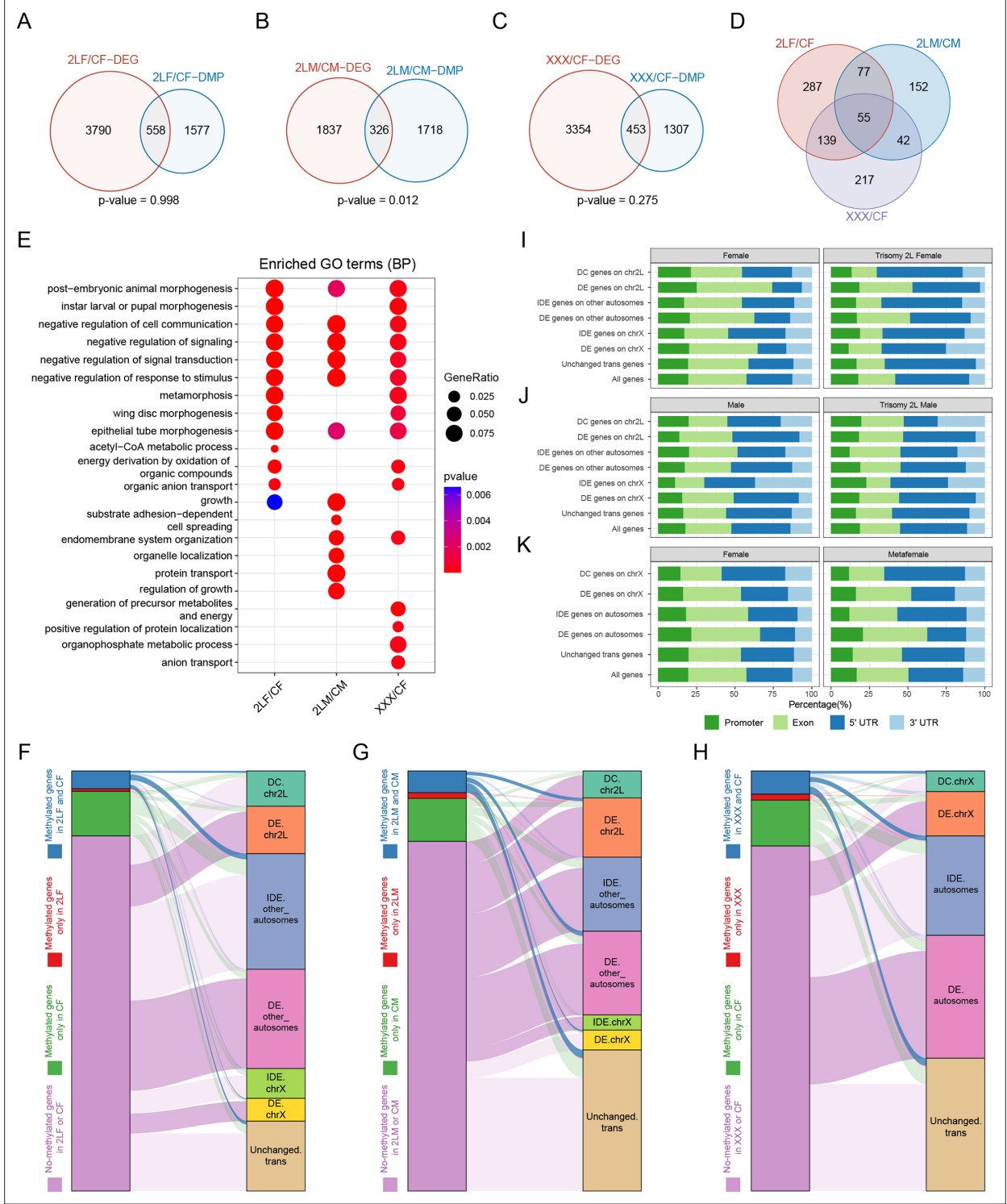

**Figure 5.** Relationships between RNA m⁶A methylation and gene expression in aneuploidy. (A–C) Venn diagrams showing the intersections of differentially expressed genes (DEGs) and differentially methylated peak (DMP) associated genes in trisomy 2L females (**A**), trisomy 2L males (**B**), and metafemales (**C**) compared with their corresponding controls. p-Values indicate one-tailed Fisher's exact tests. (**D**) The common differentially expressed and differentially methylated genes in all groups. (**E**) Functional enrichment analysis of simultaneously differentially expressed and differentially methylated genes. Top 10 enriched GO terms (Biological Process) with p-value<0.1 in each comparison were shown. (**F–H**) Sankey diagrams showing the relationships between genes with different m⁶A-modified states and genes with canonical dosage effect (DE), dosage compensation (DC), and inverse dosage effect (IDE) in trisomy 2L females (**F**), trisomy 2L males (**G**), and metafemales (**H**). Enrichment analysis was performed on each two groups of genes, and deep color lines indicate significant connection relationships (Fisher's exact test p-value<0.05). (**I–K**) Gene feature distributions for m⁶A peaks

*Figure 5 continued on next page*

*Figure 5 continued*

on genes with canonical DE, DC, and IDE in trisomy 2L females (**I**), trisomy 2L males (**J**), metafemales (**K**), and their corresponding controls. CF, wildtype female control; CM, wildtype male control; 2LF, trisomy 2L female; 2LM, trisomy 2L male; XXX, metafemale; DEG, differentially expressed gene; DMP, differentially methylated peak; DE, dosage effect; DC, dosage compensation; IDE, inverse dosage effect. Canonical DE refers to ratio >1.25, DC stands for 0.8<ratio<1.25, IDE stands for 0.5<ratio<0.8, and unchanged refers to 0.8<ratio<1.25.

The online version of this article includes the following figure supplement(s) for figure 5:

**Figure supplement 1.** Dosage-sensitive modifiers and their interactors in aneuploid *Drosophila*.

female, with significant enrichment of *cis* dosage compensation genes, autosomal *trans* dosage effect genes, and *trans* unchanged genes in m⁶A group where both trisomy and control genes are methylated; and the group without methylation is enriched for *cis* dosage effect genes and *trans* dosage effect genes (*Figure 5H*). These results indicated that m⁶A-modified genes in aneuploid *Drosophila* females are mainly related to dosage compensation and inverse dosage effect, while genes not modified by m⁶A are mainly related to direct dosage effect. Male aneuploids did not follow this trend.

Furthermore, the distributions of m⁶A modification sites along gene features in genes with classical dosage-related effects were also studied (*Figure 5I–K*). We found that in wildtype females, trisomy 2L females, and metafemales, a higher proportion of m⁶A sites on genes with canonical dosage compensation and inverse dosage effect are distributed in the 5'UTR than genes with dosage effects (*Figure 5I and K*). At the same time, this proportion is higher in trisomy than in wildtype. In contrast to females, trisomy 2L males have a higher proportion of 5'UTR RNA m⁶A modification on dosage effect genes (*Figure 5J*). The above study again demonstrates the sexual dimorphism of RNA m⁶A modification in response to aneuploidy. In addition, it has been suggested that m⁶A residues in the 5'UTR region may have unique regulatory functions (*Luo et al., 2014*; *Meyer and Jaffrey, 2014*). Therefore, we hypothesized that the high level of 5'UTR m⁶A modification on genes with dosage compensation and inverse dosage effect might regulate the down-regulation of these genes.

Dozens of dosage-sensitive modifiers have been identified in *Drosophila*, whose dosage changes can negatively or positively regulate the gene expression across the whole genome like aneuploidy (*Birchler et al., 2001*; *Birchler and Veitia, 2007*). Protein-protein interaction (PPI) networks between dosage-sensitive modifiers and differentially expressed genes (DEGs) were constructed to investigate whether RNA m⁶A modification could participate in the gene regulatory networks in unbalanced genomes through these regulators (*Figure 5—figure supplement 1A–C*). In the three types of aneuploidies, there are more dosage-sensitive modifiers with up-regulated DMPs than with down-regulated DMPs. The regulators *ox* and *Vha55* without DMPs interact with a large number of significantly up-regulated DEGs in trisomy 2L females and metafemales, whereas these two regulators interact with a small number of down-regulated DEGs in trisomy 2L males. Up-regulated DMP-associated regulators *wg*, *Uba1*, *ap*, *Atg1*, *osa*, *rdx*, and *sd* in trisomy 2L females mainly interacted with significantly down-regulated DEGs (*Figure 5—figure supplement 1A*); and up-regulated DMP-associated regulators *Uba1*, *osa*, and *sd* in metafemales also interacted with down-regulated DEGs (*Figure 5—figure supplement 1C*). Up-regulated DMP-associated regulators *wg*, *Kr-h1*, and *Trl* mainly interacted with up-regulated DEGs in trisomy 2L males (*Figure 5—figure supplement 1B*). In addition, a preponderance of significantly down-regulated DEGs interacted with dosage-sensitive modifiers is observed on *trans* chromosomes of all *Drosophila* trisomies (*Figure 5—figure supplement 1A–C*).

## Alterations of m⁶A may be involved in differential alternative splicing in imbalanced genomes

Next, we analyzed the differential alternative splicing events in aneuploid *Drosophila*. More than 1000 differential splicing events have been identified in different aneuploids, and about one-third of them are of the type of skipped exon (SE) (*Figure 6—figure supplement 1A–C*). The biological functions of differentially spliced transcripts are mainly involved in macromolecular fiber organization, locomotion, and growth (*Figure 6—figure supplement 1D*), and these genes are enriched in heterogeneous pathways in different aneuploidies (*Figure 6—figure supplement 1E*). Notably, we found that genes with DMPs are significantly enriched for differential alternative splicing events in three kinds of aneuploid *Drosophila* (Fisher's exact test p-values<0.05; *Figure 6A–C*). Among the genes whose transcripts are differentially spliced, 27–32% are also differentially methylated (*Figure 6A–C*). There

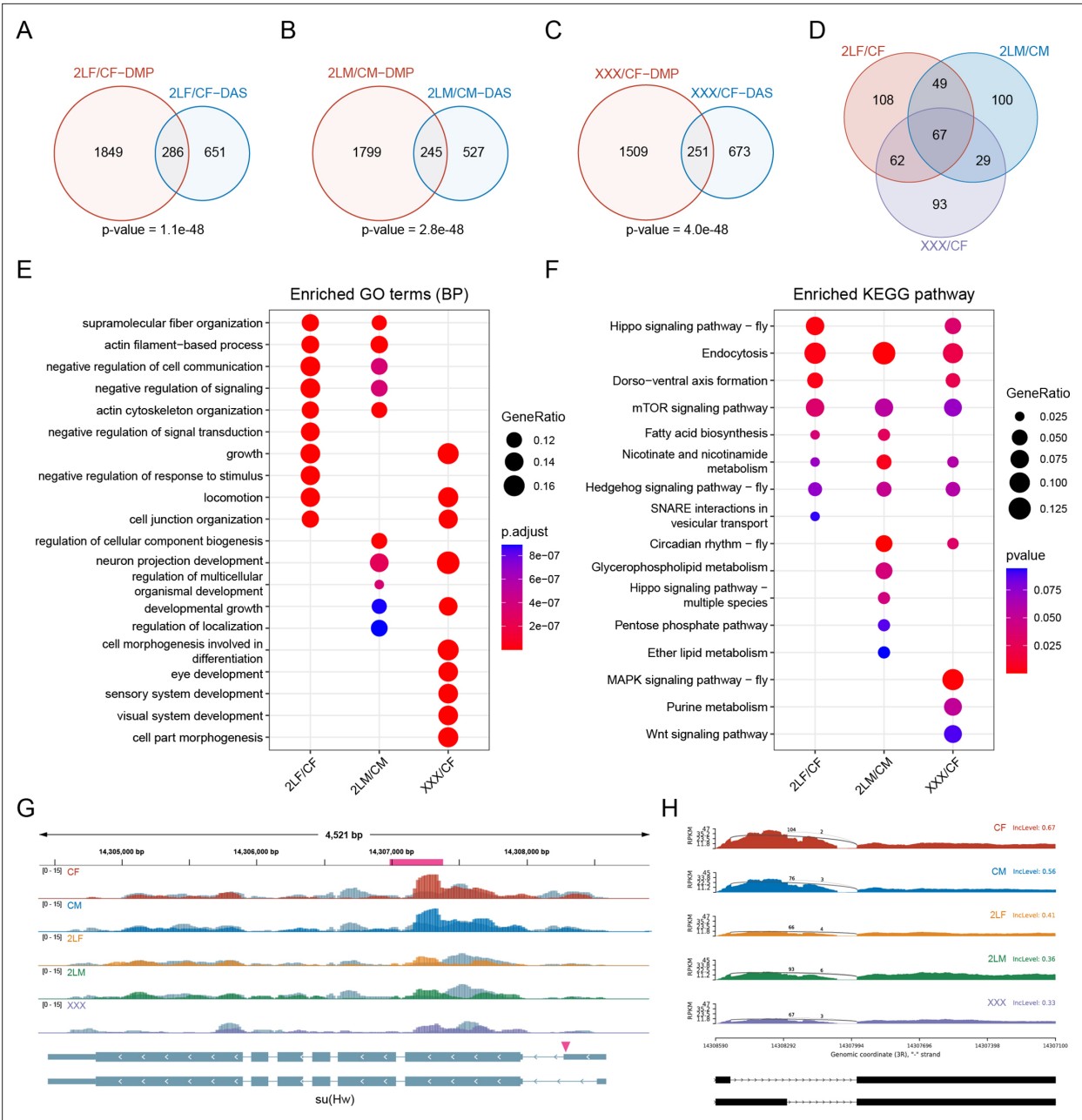

**Figure 6.** Combined analysis of differential alternative splicing (DAS) and differential methylation. (**A–C**) Venn diagrams showing the intersections of DAS genes and differentially methylated peak (DMP) associated genes in trisomy 2L females (**A**), trisomy 2L males (**B**), and metafemales (**C**) compared with their corresponding controls. p-Values indicate one-tailed Fisher's exact tests. (**D**) The common differentially alternatively spliced and differentially methylated genes in all groups. (**E**) Functional enrichment analysis of simultaneously differentially alternatively spliced and differentially methylated genes. Top 10 enriched GO terms (Biological Process) with p-value<0.05 in each comparison were shown. (**F**) Kyoto Encyclopedia of Genes and Genomes (KEGG) pathway enrichment analysis of simultaneously differentially alternatively spliced and differentially methylated genes. Top 10 enriched pathways with p-value<0.1 in each comparison were shown. (**G**) Genome browser example of *su(Hw)* for indicated m⁶A methylated RNA immunoprecipitation sequencing (MeRIP-Seq) data. Steelblue color represents input reads, while other colors represent IP reads. Signals were displayed as the mean CPM of two biological replicates. The gene architecture was shown at the bottom (only two representative transcript isoforms were shown). The magenta rectangle at above represents DMP. The magenta arrowhead indicates the position of differential alternative splicing. (**H**) Sashimi plot depicting RNA sequencing reads and exon junction reads at the position where the differential splicing events occur on *su(Hw)*. The gene model was shown below. One of the biological replicates was chosen for representation. CF, wildtype female control; CM, wildtype male control; 2LF, trisomy 2L female; 2LM, trisomy 2L male; XXX, metafemale; DAS, differential alternative splicing; DMP, differentially methylated peak; CPM, counts per million; RPKM, reads per kilobase per million mapped reads.

The online version of this article includes the following figure supplement(s) for figure 6:

*Figure 6 continued on next page*

*Figure 6 continued*

**Figure supplement 1.** Alternative splicing analysis in aneuploidies.

are 67 genes with both differential alternative splicing and differential m⁶A methylation in all aneuploi-dies (*Figure 6D*). The functions of these genes are similar to those of all differentially spliced genes (*Figure 6E*), but more consistent Kyoto Encyclopedia of Genes and Genomes (KEGG) pathways are enriched, including endocytosis, mTOR signaling pathway, Hedgehog signaling pathway, and so on (*Figure 6F*).

In all trisomy *Drosophila*, 10 genes are shown to be differentially expressed, with their transcripts also being differentially spliced and differentially methylated. Three of them are illustrated below [*su(Hw)*, *Ppn*, and *CG13124*]. *su(Hw)* is a component of the gypsy chromatin insulator complex, which is a regulatory element that establishes independent domains of transcriptional activity (*Roseman et al., 1993*). Due to the close relationship between *su(Hw)* and second-site modifiers (*Rabinow and Birchler, 1989*), and *BEAF-32*, which is also an insulator DNA-binding protein, has been proposed as a possible inverse dosage regulator (*Gurudatta et al., 2012*; *Zhang et al., 2021b*). We speculated that the transcription factor *su(Hw)* may also be a dosage-sensitive regulator. The data showed that the transcription levels of *su(Hw)* are up-regulated in all three aneuploidies, and there is a consistent m⁶A modification site with significantly down-regulated methylation (*Figure 6G*). At the same time, the transcripts of this gene have a common alternative 5' splice site (A5SS), and its inclusion levels in triso-mies are down-regulated, i.e., more short transcript isoforms are generated (*Figure 6G*). *Ppn* gene encodes an essential extracellular matrix protein that influences cell rearrangements. The expression level of *Ppn* and its m⁶A methylation in 5'UTR region are both significantly up-regulated in triso-mies (*Figure 6—figure supplement 1F*). In addition, an alternatively spliced exon is significantly less frequently skipped in all aneuploid *Drosophila* (*Figure 6—figure supplement 1G*). The third gene, *CG13124*, which may be involved in regulation of translational initiation, is up-regulated in aneu-ploids. Two m⁶A sites in its 5'UTR region are methylated at higher levels in all aneuploid *Drosophila* (*Figure 6—figure supplement 1H*). Its transcripts also have a common significantly different exon-skipping event (*Figure 6—figure supplement 1I*).

These results indicate that there are complicated relationships among RNA m⁶A modification, gene expression, and alternative splicing under the condition of genome imbalance. RNA splicing seems to be more closely related to m⁶A methylation than gene transcription. The m⁶A sites located in 5'UTR show remarkable changes in methylation levels in aneuploid *Drosophila*, and may be involved in the regulation of some differential alternative splicing events, such as exon skipping.

## Interactions between m⁶A and *Drosophila* MSL complex

Previous studies have shown that m⁶A components are involved in regulating the alternative splicing of sex-determining gene *Sxl* in *Drosophila*, and the deficiency of m⁶A writers or readers will lead to the reduction of female-specific isoforms of *Sxl* (*Haussmann et al., 2016*; *Lence et al., 2016*; *Kan et al., 2017*). *Sxl* is also a direct target of RNA m⁶A modification (*Kan et al., 2017*). We found that *Sxl* transcripts are both differentially methylated and differentially spliced in three kinds of aneuploid *Drosophila* (*Figure 7A and B*). For trisomy 2L females and metafemales, two common m⁶A peaks are significantly up-regulated in the 5'UTR region of *Sxl*. However, trisomy 2L males have a significantly down-regulated m⁶A peak in the 5'UTR (*Figure 7A*). Meanwhile, multiple junctions of *Sxl* transcripts undergo complicated alternative splicing in aneuploid *Drosophila*, including SE, A5SS, alternative 3' splice site (A3SS), and mutually exclusive exons types (*Figure 7A*). By checking the distributions of RNA sequencing (RNA-seq) reads near the male-specific exon (namely the third exon), it can be observed that there are almost no mapped reads on the third exon in wildtype females, trisomy 2L females, and metafemales, while the reads mapped to the second and fourth exons are highly prev-alent (*Figure 7B*). On the contrary, wildtype males and trisomy 2L males have a substantial number of reads on the third exon, accompanied by a smaller number of reads on the adjacent two exons (*Figure 7B*). Notably, we found that a small number of RNA-seq reads aligned to the male-specific exon appeared in trisomy 2L females, which was identified as an SE-type differential alternative splicing event (FDR = 7.4e-7; *Figure 7B*). This variation may be related to the abnormal m⁶A methyla-tion levels under the condition of genomic imbalance.

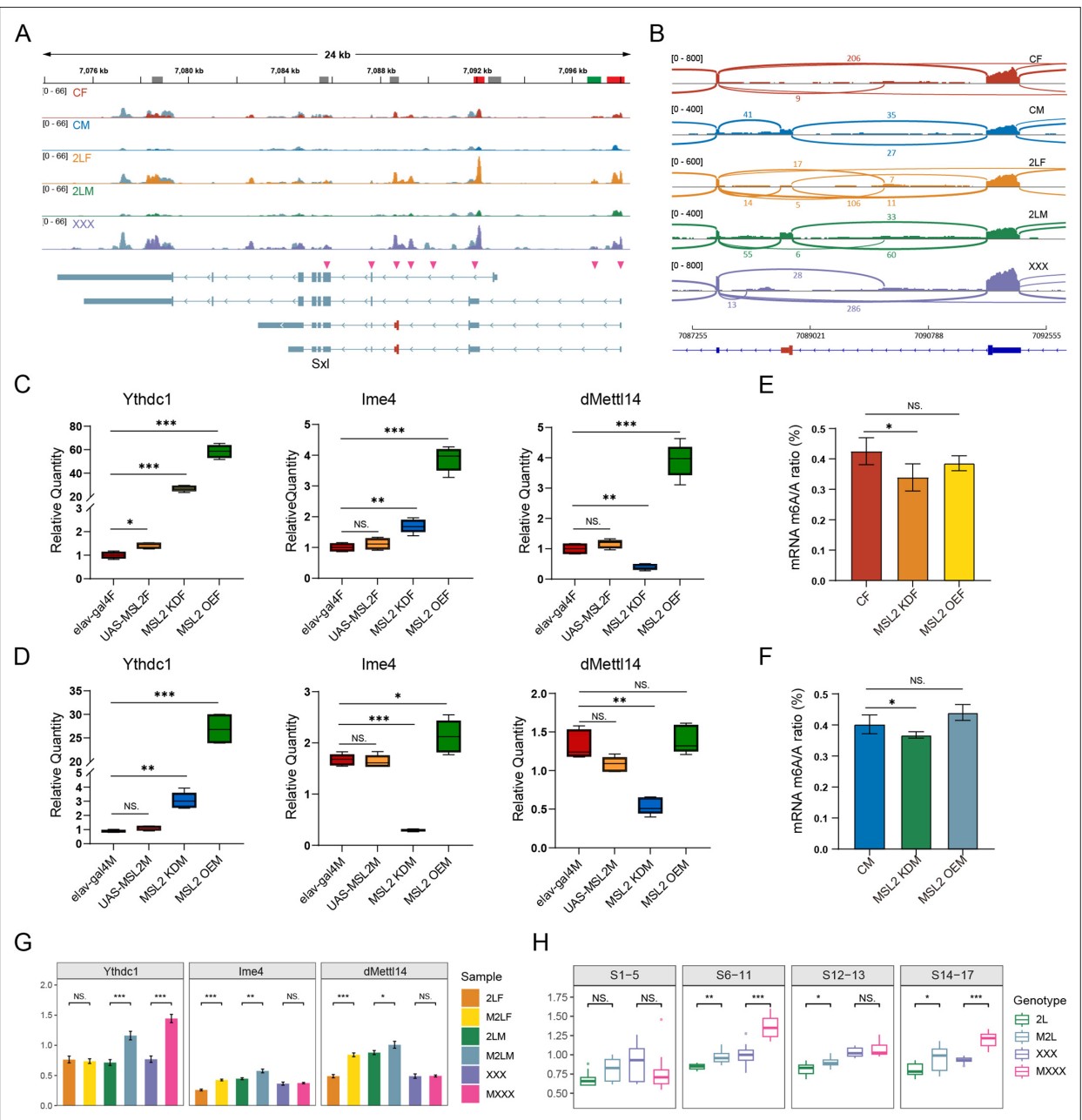

**Figure 7.** Interactions between m⁶A and *Drosophila* male-specific lethal (MSL) complex. (**A**) Genome browser example of *Sxl* for indicated m⁶A methylated RNA immunoprecipitation sequencing (MeRIP-Seq) data. Steelblue color represents input reads, while other colors represent IP reads. Signals were displayed as the mean counts per million (CPM) of two biological replicates. The gene architecture was shown at the bottom (only four representative transcript isoforms were shown). The rectangles at above represent m⁶A peaks, where red indicates up-regulated differentially methylated peaks (DMPs), green indicates down-regulated DMP, and gray indicates no significant changes. The magenta arrowheads indicate the positions of alternative splicing. (**B**) Sashimi plot depicting RNA sequencing reads and exon junction reads at the position where the differential splicing events occurs on *Sxl*. The gene model was shown below, with the third exon indicated in red. One of the biological replicates was chosen for representation. CF, wildtype female control; CM, wildtype male control; 2LF, trisomy 2L female; 2LM, trisomy 2L male; XXX, metafemale. (**C,D**) RT-qPCR analysis of messenger RNA (mRNA) levels of m⁶A components in the heads of MSL2 transgenic female (**C**) and male (**D**) *Drosophila* adults. (**E,F**) Abundance of mRNA m⁶A modification in the heads of MSL2 transgenic females (**E**) and males (**F**). MSL2 KDF, MSL2 neural-knockdown female; MSL2 KDM, MSL2 neural-knockdown male; MSL2 OEF, MSL2-overexpressed female; MSL2 OEM, MSL2-overexpressed male. Sample size = 3. Student's *t* test *p<0.05, **p<0.01, ***p<0.001. (**G**) RT-qPCR analysis of mRNA levels of m⁶A regulators in the brains of trisomy and MSL2-overexpressed trisomy *Drosophila* larvae. 2LF, trisomy 2L female; 2LM, trisomy 2L male; XXX, metafemale; M2LF, MSL2-overexpressed trisomy 2L female; M2LM, MSL2-overexpressed trisomy 2L male; MXXX, MSL2-overexpressed metafemale. Sample size = 3. Student's *t* test *p<0.05, **p<0.01, ***p<0.001. (**H**) The expression levels of Ime4 in trisomy and MSL2-overexpressed trisomy embryos represented by relative fluorescence intensity of probes. The expression of wildtype embryos

*Figure 7 continued on next page*

*Figure 7 continued*

was set as one. 2L, trisomy 2L; M2L, MSL2-overexpressed trisomy 2L; XXX, metafemale; MXXX, MSL2-overexpressed metafemale. Sample size = 10. Student's *t* test *p<0.05, **p<0.01, ***p<0.001.

The online version of this article includes the following figure supplement(s) for figure 7:

**Figure supplement 1.** The relationships between m⁶A methylation and male-specific lethal (MSL) complex.

The abnormal expression of m⁶A components reduces the survival of female *Drosophila*, which is thought to be probably caused by the expression of downstream MSL complex (*Haussmann et al., 2016*). To investigate the interplay between the MSL complex and m⁶A modification, we examined the responses of m⁶A regulators in transgenic *Drosophila* strains with MSL2 mutation or overexpression (*Figure 7—figure supplement 1A–E*; *Figure 7C and D*). The results revealed significant changes in the expression profiles of m⁶A regulators in MSL2 transgenic strains, especially the m⁶A reader protein Ythdc1, which increased tens of fold in MSL2 knockdown and overexpressed *Drosophila* (*Figure 7C and D*; *Figure 7—figure supplement 1E*). We also observed that the trends of Ime4 expression in females and males were the same in MSL2-overexpressed *Drosophila*, whereas there was obvious sexual dimorphism in MSL2-knockdown samples (*Figure 7C and D*), which may be due to the ectopic assembly of MSL complex in MSL2-overexpressed females (*Zhang et al., 2021a*). In addition, we also examined the expression levels of MSL2 when *Ythdc1* was knocked down, and found that MSL2 was also significantly increased in females (*Figure 7—figure supplement 1F*). All these results strongly suggest a potential relationship between MSL2 and Ythdc1. Next, we further compared the overall abundance of m⁶A on mRNA in MSL2 transgenic and wildtype *Drosophila* (*Figure 7E and F*). The results showed that mRNA m⁶A levels were significantly decreased with MSL2 knockdown in females and males, but overexpression of MSL2 failed to exert a discernible effect on m⁶A abundance (*Figure 7E and F*).

In the next, we investigated the expression of m⁶A regulators in aneuploid *Drosophila* overexpressing MSL2, according the results that the MSL complex could be regulated directly or indirectly by m⁶A modification, and the sexual dimorphism of RNA m⁶A modification in response to aneuploidy. It is found that the transcription levels of m⁶A regulators are significantly up-regulated in the brains of most aneuploid larvae that overexpressed MSL2 (*Figure 7G*; *Figure 7—figure supplement 1I*). These results are not completely consistent with the quantitative results when MSL2 was overexpressed in diploids, which may be related to the effect of unbalanced genomes. We also used TSA-FISH to detect the expression and distribution of m⁶A components during embryogenesis in aneuploidies overexpressing MSL2 (*Figure 7H*; *Figure 1—figure supplement 2*). The subembryonic and subcellular distributions of mRNAs for m⁶A methyltransferases and reading protein did not appear to be affected by ectopic expression of MSL2 in aneuploid *Drosophila* embryos (*Figure 1—figure supplement 2A–F*; *Figure 1—figure supplement 2A'–F'*). But the relative expression of m⁶A components showed diverse dynamics, among which the levels of the most important methyltransferase Ime4 are significantly up-regulated in autosomal trisomy and sex chromosome trisomy with MSL2 overexpression (*Figure 7H*; *Figure 1—figure supplement 2A"–F"*). Considering that unbalanced genomes can affect the expression of MSL complex subunits, and the above results show that there is a close relationship between MSL complex and m⁶A modification, we speculate that unbalanced genomes may influence the expression of m⁶A regulators, possibly through the MSL complex.

## Relationship of H4K16Ac with m⁶A modification

As an important component of the MSL complex, MOF is a histone acetyltransferase that specifically acetylates histone H4 at lysine 16 (H4K16Ac), thereby affecting chromatin structure and functions, and activating transcription (*Kind et al., 2008*; *Conrad et al., 2012*). Previous studies have elucidated the regulatory role of m⁶A in histone modifications, and histone 3 lysine 36 trimethylation (H3K36me3) also plays a role in recruiting methyltransferase complexes to deposit m⁶A markers on RNA (*Wang et al., 2018*; *Huang et al., 2019*; *Li et al., 2020*). To investigate the functions of MOF-mediated H4K16Ac on RNA m⁶A modification, we analyzed RNA-seq data from *Drosophila* strains overexpressing MOF (*Figure 8A*; *Figure 7—figure supplement 1J–L*). We found obvious changes in the expression of m⁶A regulators in strains overexpressing MOF, with the levels of almost all m⁶A regulators being elevated (*Figure 8A*). It was also observed that after overexpression of MOF, the increasing trends of

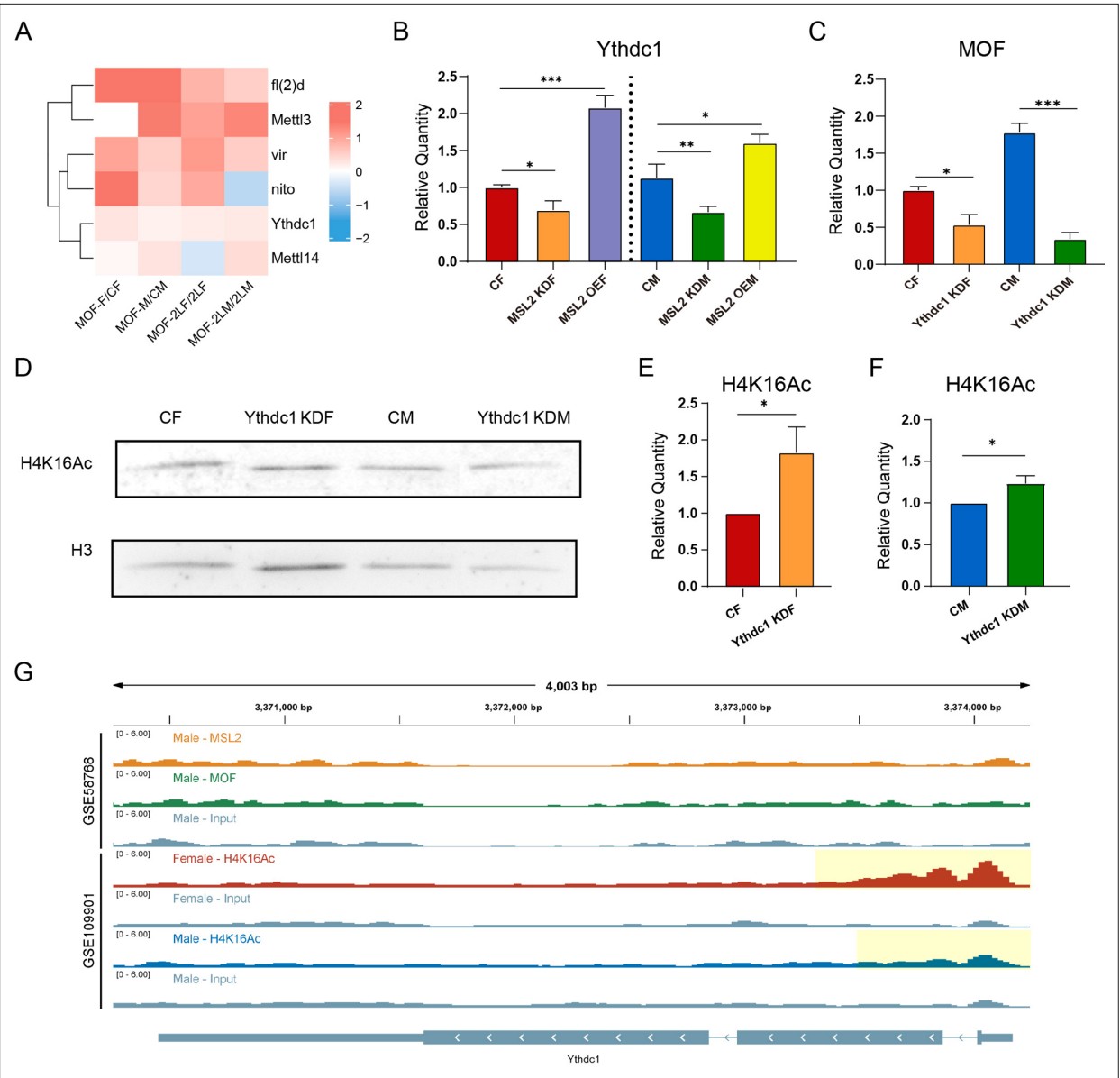

**Figure 8.** RNA m⁶A modification regulate histone acetyltransferase MOF and H4K16Ac. (**A**) Heatmap of the expression changes of m⁶A regulators in MOF overexpressing *Drosophila* larvae. The color of the heatmap represents log2(ratio). CF, wildtype female control; CM, wildtype male control; MOF-F, MOF-overexpressed female; MOF-M, MOF-overexpressed male; 2LF, trisomy 2L female; 2LM, trisomy 2L male; MOF-2LF, MOF-overexpressed trisomy 2L female; MOF-2LM, MOF-overexpressed trisomy 2L male. (**B**) RT-qPCR analysis of messenger RNA (mRNA) levels of *Ythdc1* in the heads of MSL2 transgenic *Drosophila*. MOF KDF, MOF neural-knockdown female; MOF KDM, MOF neural-knockdown male; MOF OEF, MOF-overexpressed female; MOF OEM, MOF-overexpressed male. Sample size = 3. Student's *t* test *p<0.05, **p<0.01, ***p<0.001. (**C**) RT-qPCR analysis of mRNA levels of MOF in the heads of *Ythdc1* knockdown *Drosophila*. Ythdc1 KDF, Ythdc1 neural-knockdown female; Ythdc1 KDM, Ythdc1 neural-knockdown male. Sample size = 3. Student's *t* test *p<0.05, **p<0.01, ***p<0.001. (**D**) Western blot analysis of H4K16Ac in *Drosophila*. (**E,F**) Relative quantification of H4K16Ac in wildtype and *Ythdc1* knockdown adult *Drosophila* based on western blot. Student's *t* test * Ythdc1 KDF, Ythdc1-knockdown female; Ythdc1 KDM, Ythdc1-knockdown male. Sample size = 5. Student's *t* test *p<0.05. (**G**) Genome browser example of *Ythdc1* for indicated ChIP-seq data. Signals were displayed as counts per million (CPM) values. The gene architecture was shown at the bottom (only one representative transcript isoform was shown). The yellow shaded area indicates the presence of the H4K16Ac peaks.

The online version of this article includes the following source data and figure supplement(s) for figure 8:

**Source data 1.** PDF file containing original western blots for *Figure 8D*, indicating the relevant bands and treatments.

**Source data 2.** Original files for western blot analysis displayed in *Figure 8D*.

**Figure supplement 1.** Ythdc1 may regulate H4K16Ac through histone acetyltransferase MOF.

**Figure supplement 2.** Genome browser example of m⁶A regulator genes for indicated ChIP-seq data.

m⁶A regulators in females and males were not exactly the same, mostly showing more pronounced increases in females, except for Mettl3 (Ime4) and dMettl14. According to the results of RT-qPCR, the expression of Ythdc1 was decreased in MOF knockdown strains and increased in MOF overexpression strains (*Figure 8B*; *Figure 7—figure supplement 1J*); meanwhile, the expression levels of MOF were significantly reduced in *Ythdc1* neural-knockdown strains (*Figure 8C*). These results were further verified by polytene chromosome immunofluorescence experiments (*Figure 8—figure supplement 1E and F*). In males, the expression level of MOF was significantly decreased in *Ythdc1* neural-knockdown *Drosophila*; while in *Ythdc1* neural-knockdown females, the expression level of MOF was not significantly changed compared with wildtype females (*Figure 8—figure supplement 1E and F*). These results suggest that MOF-mediated acetylation modification can have a certain effect on RNA m⁶A methylation in a sexually dimorphic manner, which may be due to the endogenous MSL complex and the imbalance of X chromosome dosage in males.

Next, we investigated whether the m⁶A reader protein Ythdc1 would also have an effect on the levels of H4K16Ac in *Drosophila*. To this end, we employed western blot analysis to assess the expression patterns of H4K16Ac in *Ythdc1* knockdown *Drosophila* adults (*Figure 8D*). The subsequent quantitative analysis revealed a significant up-regulation of H4K16Ac in both female and male (*Figure 8E and F*). Furthermore, to substantiate our findings, we conducted polytene chromosome immunofluorescence in third instar larvae (*Figure 8—figure supplement 1G*). Quantitative analysis of these assays in *Ythdc1* knockdown *Drosophila* showed that the changes of H4K16Ac levels also showed sexual dimorphism (*Figure 8—figure supplement 1H*). In males, knockdown of *Ythdc1* led to a decrease in H4K16Ac level; whereas in females, knockdown of *Ythdc1* did not affect the level of H4K16Ac (*Figure 8—figure supplement 1H*). These observed changes in H4K16Ac were consistent with the pattern of changes in MOF protein in *Ythdc1* knockdown *Drosophila* strains, suggesting that m⁶A modification may affect H4K16Ac levels through the mediation of MOF. Overall, while the alterations in H4K16Ac levels are not uniformly consistent between larvae and adult *Drosophila*, these findings nonetheless demonstrate that the knockdown of *Ythdc1* has a significant impact on the expression levels of H4K16Ac at different stages of development and imply a potential relationship between H4K16Ac with m⁶A modification.

We analyzed two ChIP-seq datasets (GSE109901 and GSE58768) to study whether m⁶A regulator genes (especially *Ythdc1*) are targets of DCC components and H4K16Ac. According to the results, most of the m⁶A regulator genes, including *Ythdc1*, contain H4K16Ac peaks in both sexes, all of which are located in the 5' regions (*Figure 8G*; *Figure 8—figure supplement 2*); except that *Ime4* shows sexual dimorphism and only contains H4K16Ac peak in females. On the other hand, analysis of ChIP-seq data of MSL2 and MOF in male *Drosophila* showed that most of MSL2 and MOF peaks were located on the X chromosome (99.1% of MSL2 peaks and 61.6% of MOF peaks), which may be due to the fact that MSL2 and MOF are mostly tethered to the X chromosome by MSL complex under physiological conditions (*Bashaw and Baker, 1995*; *Kelley et al., 1995*; *Kind et al., 2008*; *Conrad et al., 2012*). Therefore, there is no MSL2 and MOF peak near the m⁶A regulator genes which located on the autosomes (*Figure 8H*; *Figure 8—figure supplement 2*). These results showed that there is a direct relationship between m⁶A regulators and H4K16Ac, but there is no evidence that m⁶A regulator genes are direct targets of DCC components. MSL2 and MOF may thereby interact with m⁶A regulators in other ways.

To further study whether unbalanced genomes are involved in the interaction between m⁶A and histone acetylation modification, we also analyzed RNA-seq data from trisomy 2L *Drosophila* strains overexpressing MOF. The data showed that overexpression of MOF in trisomy 2L resulted in significant changes in the expression of m⁶A regulators, with a trend different from that observed in diploids (*Figure 8A*). These results suggest that genomic imbalance might affect the interaction between MOF and m⁶A regulators to some extent, and the potential mechanisms require further investigation.

## Discussion

As an emerging epigenetic modification, RNA m⁶A methylation has been found to be involved in almost all aspects of RNA fate and metabolism (*Gilbert et al., 2016*; *Yang et al., 2018*). RNA m⁶A modification is also closely related to the development of organisms and a variety of human diseases (*Barbieri et al., 2017*; *Zhao et al., 2017*; *Pinello et al., 2018*; *Ma et al., 2019*; *Liu et al., 2021*; *Shafik et al., 2021*). However, the roles of m⁶A methylation in development and gene expression of

aneuploidy have not been studied yet. Aneuploid variation is usually more detrimental than changes of the entire chromosome set due to genomic imbalance (*Birchler and Veitia, 2007*; *Birchler and Veitia, 2012*). The global changes of gene regulatory networks in unbalanced genomes involves various epigenetic mechanisms, including histone modification, chromatin remodeling, lncRNAs, microRNAs, etc. (*Birchler, 2016*; *Zhang et al., 2021a*; *Zhang et al., 2023*; *Shi et al., 2022*). This study demonstrated that the expression of m$^6$A components was altered under genomic imbalance, leading to dynamic changes in the entire methylome. Potential intermediaries by which m$^6$A modification could affect *trans* regulation and achieve dosage compensation in aneuploid *Drosophila* were also investigated, such as dosage-sensitive modifiers, alternative splicing events, and the MSL complex.

Our experiments show that the expression levels of most m$^6$A component genes are significantly down-regulated in aneuploid *Drosophila* larvae (*Figure 1A*; *Figure 1—figure supplement 1C*). Depletion of m$^6$A components interferes with the development of animals and plants, especially leading to impaired self-renewal and differentiation of embryonic stem cells, defects in embryonic development, and even early embryonic lethality (*Granadino et al., 1990*; *Horiuchi et al., 2006*; *Raffel et al., 2007*; *Luo et al., 2014*; *Wang et al., 2014a*; *Geula et al., 2015*). We demonstrated by TSA-FISH that appropriate temporal and spatial specific distributions of the transcripts for m$^6$A components are vital during *Drosophila* embryogenesis (*Figure 1C–F*; *Figure 1—figure supplement 2*). The abnormal expression of m$^6$A-related genes may affect the development of aneuploid embryos.

In previous studies, the abundance of m$^6$A modification is usually positively correlated with the number of m$^6$A peaks and m$^6$A-marked genes (*Luo et al., 2014*; *Zhu et al., 2023*). However, our results obviously did not conform to this rule, with higher m$^6$A abundance and fewer MeRIP-Seq peaks in aneuploids (*Figure 2A–C*). This reflects the complexity and heterogeneity of m$^6$A modification. We suspect that in aneuploidy many RNAs may be lost that are methylated at a low level in wildtype, and possess a higher proportion of highly methylated RNAs. Analysis of the expression levels at each m$^6$A site confirmed our hypothesis (*Figure 2D*). Thus, this phenomenon represents an imbalance in m$^6$A methylation caused by aneuploidy. In addition, it is worth noting that due to the limitation of the larval samples, our detection of the overall abundance of m$^6$A in aneuploidy is carried out for total RNA, including all types of RNA such as mRNA, lncRNA, and rRNA, and may be slightly different from detection of mRNA only. However, according to the results of m$^6$A abundance detection in *Drosophila* adult heads, the enrichment or non-enrichment of mRNA from total RNA did not make a substantive difference in the results.

The distribution of m$^6$A sites on gene features, m$^6$A consensus motifs, biotypes of m$^6$A-marked genes, enriched functions and pathways of methylated genes in aneuploidies are similar to those in wildtype. We also found that genes highly methylated in all genotypes are enriched for transcription factors (*Figure 2I*). It is consistent with previous studies suggesting that transcription regulatory genes may be preferentially targeted by m$^6$A (*Kan et al., 2017*).

We also analyzed the characteristics of DMPs and DMP-associated genes in trisomies. By observing the distributions of MeRIP-Seq reads around DMP sites, it can be found that the densities of m$^6$A-marked reads on chromosome 2L are relatively higher in trisomy 2L females and males (*Figure 3F and G*). Meanwhile, metafemales with triple X have a higher density of m$^6$A-marked reads on chromosome X (*Figure 3H*). Previous studies have found that there is dosage compensation for *cis* genes in autosomal and sex chromosome aneuploid *Drosophila*, and the expression of most genes on varied chromosomes approaches diploid levels (*Sun et al., 2013b*; *Sun et al., 2013c*). Therefore, the up-regulation of m$^6$A levels is not directly caused by the increased number of chromosomes. A recent study found that the selective enrichment of m$^6$A methylation may play a role as a transcript degradation signal in dosage compensation in mammals (*Rücklé et al., 2023*). Therefore, we speculate that changes in m$^6$A levels in aneuploid *Drosophila* may affect its dosage compensation and inverse dosage effect by regulating the stability of the transcripts. In addition, more up-regulated DMPs are detected in trisomy 2L males and metafemales, while trisomy 2L females have more down-regulated DMPs (*Figure 3A–D*). Combined with the fact that the survival rate of trisomy 2L female larvae is relatively lower than that of the other two trisomies, it can be speculated that the regulation of DMPs may affect the survival and development of aneuploid *Drosophila*.

Gene expression in unbalanced genomes is extensively modulated (*Sun et al., 2013b*; *Sun et al., 2013c*; *Hou et al., 2018*; *Raznahan et al., 2018*; *Shi et al., 2021*; *Yang et al., 2021*; *Zhang et al., 2021b*; *San Roman et al., 2023*). Previous studies have analyzed the transcriptome data of autosomal

and sex chromosome trisomic *Drosophila*, and found that most of the genes on the triple chromosomes were compensated, and the ratios of their expression levels to wildtype were approximately 1; meanwhile, the expression of genes on other chromosomes was close to two-thirds of that of the wildtype, which was called an inverse dosage effect (*Sun et al., 2013b*; *Sun et al., 2013c*; *Zhang et al., 2021b*). We investigated the relationships between genes with different m⁶A methylation status and genes modulated by classical dosage-related effects (*Figure 5F–H*). The results showed that for aneuploid females, genes methylated in both trisomy and control are significantly enriched in dosage compensated *cis* genes and inverse dosage effect *trans* genes, and genes not methylated at all are enriched in dosage effect genes. However, in aneuploid males, methylated genes are associated with gene dosage effect (*Figure 5F–H*). We also found that the proportion of 5'UTR m⁶A peaks on dosage compensation and inverse dosage effect genes are increased in aneuploid females (*Figure 5I–K*). These results provide evidence of sexual dimorphism in the relationships between RNA m⁶A modification and dosage-related effects. m⁶A located in the 5'UTR generally shows higher tissue specificity and richer dynamic changes, and may have unique regulatory functions (*Dominissini et al., 2012*; *Meyer and Jaffrey, 2014*; *Gilbert et al., 2016*). The increased proportion of m⁶A in the 5'UTR of genes with classical dosage-related effects in aneuploidies suggests that m⁶A modification may be involved in the regulation of dosage-dependent genes.

Dosage-related effects of aneuploidy are thought to be caused by dosage-sensitive genes (*Shi et al., 2021*; *Yang et al., 2021*). Some dosage-sensitive regulators have been identified in *Drosophila*, and changes in the dosage of individual regulatory genes can mimic the effects of aneuploidy in the whole genome (*Birchler et al., 2001*; *Xie and Birchler, 2012*; *Zhang et al., 2021b*). Most dosage-sensitive regulators are transcription factors, signal transduction components, and chromatin proteins, which have in common being members of macromolecular complexes or having multicomponent interactions (*Birchler et al., 2001*; *Birchler and Veitia, 2007*). We studied the m⁶A modification of dosage-dependent genes and their PPI networks in aneuploid *Drosophila* (*Figure 5—figure supplement 1*). The results showed that there are complex interactions between dosage-sensitive regulators and differentially expressed genes. Among them, most of the DEGs located on the unvaried chromosomes are down-regulated, indicating that the dosage-sensitive regulators mainly have negative effects on *trans* target genes. There are more regulatory genes with up-regulated DMPs than down-regulated DMPs in all three aneuploidies, and many of the regulators with up-regulated methylation are connected with interactors that have down-regulated expression.

We comprehensively analyzed the relationships among RNA m⁶A modification, gene expression levels, and alternative splicing. The data showed that differential m⁶A methylation under genomic imbalance appeared to be more closely associated with differential alternative splicing (*Figure 5A–C*; *Figure 6A–C*). This phenomenon is reasonable because the mutation or knockout of *Ime4*, *dMettl14*, *fl(2)d*, and *Ythdc1* has been found to affect a large number of alternative splicing events in *Drosophila*, and *fl(2)d* itself is thought to encode a splicing factor (*Penn et al., 2008*; *Haussmann et al., 2016*; *Lence et al., 2016*). A small set of transcripts are simultaneously differentially m⁶A methylated, differentially expressed, and differentially alternative spliced in aneuploidies, including the transcription factor *su(Hw)*, which may be a dosage-sensitive regulator (*Figure 6G–H*). Besides, m⁶A modification in the 5'UTR regions may play a special role in some differential alternative splicing events (*Figure 6—figure supplement 1F–I*).

RNA m⁶A modification in *Drosophila* has been shown to be involved in the alternative splicing of *Sxl* (*Lence et al., 2017*). The deletion of some m⁶A components in females will result in a reduction of female-specific splicing of *Sxl* and an increase of the inclusion of the third exon, along with phenotypic sexual transformation (*Haussmann et al., 2016*; *Lence et al., 2016*; *Kan et al., 2017*). We found that *Sxl* transcripts of aneuploid *Drosophila* are both differentially m⁶A methylated and differentially spliced (*Figure 7A and B*). Among multiple alternative splicing events, trisomy 2L females have a higher level of the third exon inclusion compared with wildtype females (*Figure 7B*), which may be related to the differential methylation at 5'UTR of the *Sxl* transcripts.

Some studies proposed that the deletion of RNA m⁶A methyltransferase Ime4 harms the survival of female *Drosophila* because of the insufficient inhibition of *msl-2* caused by decreased *Sxl* levels, which in turn leads to up-regulation of X-linked genes (*Haussmann et al., 2016*). However, other studies have found that the expression of genes on the X chromosome in females with ectopic expression of MSL2 does not increase twofold, i.e., the MSL complex does not directly mediate dosage

compensation, and the global inverse dosage effect caused by the imbalance of sex chromosomes is the basis for dosage compensation (**Birchler, 1981**; **Birchler, 2016**; **Sun et al., 2013a**). Therefore, the lethality of the lack of m⁶A in female *Drosophila* may not be directly caused by up-regulation of X-linked genes through ectopic assembly of MSL complex, and the specific reasons need to be further studied.

The functions of MSL complex in dynamic regulation of global gene expression in aneuploid genomes have been described (**Zhang et al., 2021a**). Here, we found that MSL2, a structure component of MSL complex, not only affects the expression of m⁶A regulators, but also influences the overall abundance of mRNA m⁶A in *Drosophila* (**Figure 7C–F**), proving a close relationship between the MSL complex and RNA m⁶A modification. In addition, under the condition of genomic imbalance, the relative expression of m⁶A components was also changed in larvae and embryos of MSL2-overexpressed trisomies (**Figure 7G and H**). Another component of the MSL complex, MOF, which mediates histone acetylation and transcriptional activation (**Kind et al., 2008**; **Conrad et al., 2012**), is also closely associated with the m⁶A reader Ythdc1. Overexpression of MOF increased the expression of Ythdc1 (**Figure 8A and B**), and in turn, knockdown of *Ythdc1* influenced the expression of MOF and the level of H4K16Ac catalyzed by it (**Figure 8C–F**; **Figure 8—figure supplement 1G and H**). It is worth noting that the rationale behind the variable expression levels of H4K16Ac in *Ythdc1* knockdown *Drosophila* across different developmental stages merits further investigation. These results demonstrate complicated interactions between RNA m⁶A methylation and the MSL complex in unbalanced genomes, which may affect the gene expression, sexual dimorphism, development, and survival of aneuploid *Drosophila*.

## Materials and methods
### *Drosophila* stocks and genetic crosses
The *Drosophila* strains mentioned in this study were all maintained and crossed in our laboratory. The crossing methods have been described previously (**Zhang et al., 2021a**). Trisomy chromosome 2 left arm (2L) female and male third instar larvae were obtained from the cross of *y; C(2L)dp; F(2R) bw* females and Canton S males. The metafemale larvae were obtained from the cross of *C(1)DX, ywf/winscy* females and Canton S males. Aneuploidies overexpressing MSL2 were generated from the crosses of *Drosophila* with compound chromosomes and MSL2 homozygotes. The MSL2 transgene strain was constructed and validated in a previous study (**Sun et al., 2013a**). All *Drosophila* strains were cultured on cornmeal dextrose medium at 25°C. Genes and chromosomal balancers are described in Flybase (https://flybase.org/).

### RNA m⁶A methylation quantification
Total RNA was extracted from *Drosophila* larvae using TRIzol Reagent (Invitrogen) and mRNA was isolated using the Dynabeads mRNA purification kit (Invitrogen, 61006) to detect the abundance of m⁶A. The relative quantification of m⁶A methylation was performed using EpiQuik m⁶A RNA Methylation Quantification Kit (Colorimetric) (Epigentek, NY, USA, Cat # P-9005). Specifically, 80 µl of Binding Solution was first added to each well of the plate, and 200 ng of total RNA or mRNA samples, 2 µl of Negative Control, or 2 µl of diluted Positive Control were added to the designed wells. Subsequently, the plate was incubated at 37°C for 90 min. After washing with Wash Buffer, 50 µl of Capture Antibody, 50 µl of Detection Antibody, and 50 µl of Enhancer Solution were added to each well in order, and wells were emptied before adding a new solution each time. Finally, 100 µl of Developer Solution and Stop Solution were added to each well away from light, and the absorbance was read on a microplate reader at a wavelength of 450 nm.

### m⁶A methylated RNA immunoprecipitation sequencing
The MeRIP-Seq service was provided by CloudSeq Biotech Inc (Shanghai, China), and this technology was developed on the basis of published experimental methods (**Meyer et al., 2012**). In brief, *Drosophila* larvae of five genotypes, each with two biological replicates, were collected for sequencing. Ribosomal RNAs were removed from total RNA using Ribo-Zero rRNA Removal Kits (Illumina, USA). Immunoprecipitation of m⁶A RNA was performed using GenSeq m⁶A RNA IP kit (GenSeq, Shanghai, China). The NEBNext Ultra II Directional RNA Library Prep kit (New England Biolabs, USA)

was used for RNA-seq library construction. High-throughput sequencing was performed using Illumina NovaSeq 6000 sequencers with the paired-end 150 bp protocol.

## Analysis of MeRIP-Seq data

The raw sequencing data was first filtered by Trim Galore (version 0.6.10) (https://www.bioinformatics.babraham.ac.uk/projects/trim_galore/) to remove adapters and low-quality reads. The quality of the data was then assessed by FastQC (version 0.12.1) (https://www.bioinformatics.babraham.ac.uk/projects/fastqc/). Subsequently, clean reads were aligned to the *Drosophila* reference genome ( Drosophila_melanogaster.BDGP6.32.dna.toplevel.fa, downloaded from the Ensembl database) using HISAT2 (version 2.2.1) (*Kim et al., 2019*). Next, The R package exomePeak2 (version 1.10.0) (*Meng et al., 2013*) was used for m6A peak calling (the screening criteria were log2FC≥1, RPM.IP≥0.5, and score≥5). Motif analysis was performed using HOMER (version 4.11) (*Heinz et al., 2010*). DMPs were analyzed by the R package DiffBind (version 3.8.4) (*Stark and Brown, 2013*), which employs the DESeq2 (*Love et al., 2014*) method. m6A peaks with a p-value of 0.1 or less were considered as DMPs. ChIPseeker (version 1.34.1) (*Yu et al., 2015*) was used to annotate the peaks, and plotted some of the figures. The profiles and heatmaps of MeRIP-Seq reads around DMPs were generated by deepTools (version 3.5.4) (*Ramírez et al., 2014*). The signal distribution on the genome was visualized using the Integrative Genomics Viewer (IGV) software (version 2.12.0) (*Thorvaldsdóttir et al., 2013*).

## RNA-seq data and analysis

The RNA-seq data of aneuploid *Drosophila* used in this article were generated in a previous study (*Zhang et al., 2023*) and can be downloaded from the Gene Expression Omnibus (GEO) database (GSE233534). Genome mapping of sequencing data and gene expression quantification were carried out through HISAT2-StringTie pipeline (*Pertea et al., 2015*; *Kim et al., 2019*). Differential alternative splicing events were identified by rMATS (version 4.2.0) program (*Shen et al., 2014*) (with the parameters of -t paired `--readLength` 150 `--cstat` 0.0001 `--libType` fr-firststrand). The method of generating the ratio distributions of gene expression changes was as described previously (*Zhang et al., 2022*). The plots were generated using ggplot2 (version 3.4.0) (*Villanueva and Chen, 2019*) in the R program (version 4.2.2). Differential expression analysis was performed by DESeq2 (version 1.38.2) (*Love et al., 2014*), and the threshold was set to adjusted p-value≤0.05. The functional enrichment analysis was performed by ClusterProfiler (version 4.6.0) (*Wu et al., 2021*) based on org.Dm.eg. db (version 3.16.0) from Bioconductor (https://www.bioconductor.org/). The KEGG pathway data was obtained from the network (https://www.kegg.jp/). The list of transcription factors of *Drosophila* was downloaded from AnimalTFDB (*Hu et al., 2019*). PPI relationships were obtained from the STRING database (*Szklarczyk et al., 2019*).

## Real-time quantitative PCR

Total RNA of *Drosophila* whole larvae, larval brains, or adult heads was extracted using TRIzol Reagent (Invitrogen), and reverse transcription was done with *TransScript* one-step gDNA Removal and cDNA Synthesis SuperMix (TransGen Biotech). The sequences of primers designed for RT-qPCR were listed in *Figure 1—figure supplement 1A*. The usability of these primers was verified by agarose gel electrophoresis (*Figure 1—figure supplement 1B*). The real-time PCR was performed with *TransStart* Tip Green qPCR SuperMix (+Dye II) (TransGen Biotech) using ABI QuantStudio 6 Flex Real-Time PCR System. Relative quantification of gene expression was determined using the $2^{-\Delta\Delta Ct}$ method.

## FISH of *Drosophila* embryos

Collection of *Drosophila* embryos and FISH were performed as previously described (*Jandura et al., 2017*; *Zhang et al., 2021a*). The probe primers containing flanking T7 promoter elements were listed in *Figure 1—figure supplement 1D*. After PCR amplification and in vitro transcription, their products were examined by agarose gel electrophoresis (*Figure 1—figure supplement 1E*). Digoxygenin (DIG)-labeled antisense RNA probes were hybridized to transcripts of interest, and then detected using a succession of anti-DIG antibody conjugated to biotin, streptavidin conjugated to horseradish peroxidase (HRP) and fluorescently conjugated tyramide. This hierarchical process can greatly enhance the probe signals and is referred to as a TSA system. All images were acquired with a Zeiss LSM880 laser confocal fluorescence microscope using ZEN software. The

same probes for relative fluorescence intensity analysis were photographed using the same parameters. Fluorescence images were processed and analyzed using Fiji (version 1.53c) (*Schindelin et al., 2012*).

## Polytene chromosomes immunostaining

Salivary gland chromosomes immunostaining was performed as previously described (*Zhang et al., 2021a*). Briefly speaking, the salivary glands from third-instar larvae were first fixed in 3.7% formaldehyde for 1 min, and then dissociated with 50% acetic acid for 5 min. Polytene chromosomes were treated with the following primary antibodies at a dilution of 1:100: anti-SXL (Developmental Studies Hybridoma Bank, M18-s), anti-MSL2 (Santa Cruz, sc-32459), anti-MOF (Santa Cruz, sc-22351), and anti-H4K16Ac (EMD Millipore, 07-329). Finally, fluorescence-conjugated secondary antibodies (Alexa Fluor 488 and Alexa Fluor 594, Jackson ImmunoResearch) were used for detection.

## Histone extraction and western blotting

*Drosophila* adults were ground and lysed in Triton Extraction Buffer (PBS at pH 7.4 with 0.5% Triton X-100, 0.5 mM phenymethylsulfonyl fluoride, and 0.02% sodium butyrate) and histones were acid-extracted in 0.2 N HCl overnight. Acid-extracted histones were then run on a 12% Tris-Glycine gel and blotted onto a PVDF membrane. Antibodies for Histone H3 (NB500-171, Novus) and H4K16Ac (07-329, Sigma) were all incubated at a dilution of 1:1000 in 5% skim milk powder solution. Westerns blots were imaged and protein levels quantified using the ImageJ software.

## Acknowledgements

This work was supported by National Natural Science Foundation of China (Grant No. 32070566) to LS. We thank James A Birchler for revising the manuscript.

## Additional information

### Funding

| Funder | Grant reference number | Author |
|---|---|---|
| National Natural Science Foundation of China | 32070566 | Lin Sun |

The funders had no role in study design, data collection and interpretation, or the decision to submit the work for publication.

### Author contributions

Shuai Zhang, Data curation, Formal analysis, Investigation, Visualization, Methodology, Writing – original draft, Writing – review and editing; Ruixue Wang, Kun Luo, Validation, Investigation, Visualization, Methodology, Writing – original draft, Writing – review and editing; Shipeng Gu, Investigation, Methodology, Writing – original draft; Xinyu Liu, Validation, Investigation, Methodology, Writing – original draft; Junhan Wang, Formal analysis, Investigation, Methodology; Ludan Zhang, Validation, Investigation, Methodology; Lin Sun, Conceptualization, Formal analysis, Supervision, Funding acquisition, Investigation, Methodology, Writing – original draft, Project administration, Writing – review and editing

### Author ORCIDs

Ruixue Wang http://orcid.org/0000-0001-5046-5023
Lin Sun https://orcid.org/0000-0002-8827-8319

Reviewer #1 (Public review): https://doi.org/10.7554/eLife.100144.3.sa1
Reviewer #2 (Public review): https://doi.org/10.7554/eLife.100144.3.sa2
Author response https://doi.org/10.7554/eLife.100144.3.sa3

# Additional files

## Supplementary files
MDAR checklist

## Data availability

Sequencing data generated in this study have been deposited in GEO under accession number GSE253401.

The following dataset was generated:

| Author(s) | Year | Dataset title | Dataset URL | Database and Identifier |
|---|---|---|---|---|
| Zhang S, Liu X, Wang R, Wang J, Zhang L, Sun L | 2024 | Dynamics and regulatory roles of RNA m6A methylation in unbalanced genomes | https://www.ncbi.nlm.nih.gov/geo/query/acc.cgi?acc=GSE253401 | NCBI Gene Expression Omnibus, GSE253401 |

The following previously published datasets were used:

| Author(s) | Year | Dataset title | Dataset URL | Database and Identifier |
|---|---|---|---|---|
| Zhang S, Wang R, Zhu X, Zhang L, Liu X, Sun L | 2024 | Characteristics and expression of lncRNA in *Drosophila* aneuploidy | https://www.ncbi.nlm.nih.gov/geo/query/acc.cgi?acc=GSE233534 | NCBI Gene Expression Omnibus, GSE233534 |
| Renschler G, Richard G, Valsecchi CIK, Toscano S | 2018 | Facultative dosage compensation of developmental genes on autosomes in *Drosophila* and mammals | https://www.ncbi.nlm.nih.gov/geo/query/acc.cgi?acc=GSE109901 | NCBI Gene Expression Omnibus, GSE109901 |
| Figueiredo ML, Kim M, Philip P, Allgardsson A, Stenberg P, Larsson J | 2014 | Non-coding roX RNAs prevent the binding of the MSL-complex to heterochromatic regions | https://www.ncbi.nlm.nih.gov/geo/query/acc.cgi?acc=GSE58768 | NCBI Gene Expression Omnibus, GSE58768 |

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

# Appendix 1

**Appendix 1—key resources table**

| Reagent type (species) or resource | Designation | Source or reference | Identifiers | Additional information |
|---|---|---|---|---|
| Antibody | anti-SXL (Mouse monoclonal) | Developmental Studies Hybridoma Bank | Cat# M18-s, RRID:AB_528464 | IF(1:100) |
| Antibody | anti-MSL2 (Goat monoclonal) | Santa Cruz | Cat# sc-32459, RRID:AB_672213 | IF(1:100) |
| Antibody | anti-MOF (Goat monoclonal) | Santa Cruz | Cat# sc-22351, RRID:AB_670132 | IF(1:100) |
| Antibody | anti-H3 (Rabbit polyclonal) | Novus | Cat# NB500-171, RRID:AB_10001790 | WB(1:1000) |
| Antibody | anti-H4K16Ac (Rabbit monoclonal) | EMD Millipore | Cat# 07-329, RRID:AB_310525 | IF(1:100) WB(1:1000) |
| Antibody | anti-Digoxigenin (Mouse monoclonal) | Jackson Immuno Research | Cat# 200-062-156, RRID:AB_233901 | TSA-FISH(1:400) |
| Antibody | Streptavidin-HRP (Rabbit polyclonal) | Invitrogen | Cat# S991 | TSA-FISH(1:1000) |
| Sequence-based reagent | Ime4-L | This paper | PCR primers | CAAGTACGTGCACTATGAGG |
| Sequence-based reagent | Ime4-R | This paper | PCR primers | GTCATGTCCAAGAAGCGTAG |
| Sequence-based reagent | dMettl14-L | This paper | PCR primers | GCCTCTTCCTCCAAGAAAAC |
| Sequence-based reagent | dMettl14-R | This paper | PCR primers | CCTCAACTTGGGATACTCCT |
| Sequence-based reagent | fl(2)d-L | This paper | PCR primers | CCTGGACGTTATTTCCTACT |
| Sequence-based reagent | fl(2)d-R | This paper | PCR primers | TAAGCTAGAGACCATTCACG |
| Sequence-based reagent | vir-L | This paper | PCR primers | GTACATGAAACCCTTAGAGGC |
| Sequence-based reagent | vir-R | This paper | PCR primers | CTTGCTTATGGAGAGATAGCG |
| Sequence-based reagent | nito-L | This paper | PCR primers | GCCAGTACGGTTCCAGATGT |
| Sequence-based reagent | nito-R | This paper | PCR primers | CCGTCCGTCAAATGAAACTT |
| Sequence-based reagent | Ythdc1-L | This paper | PCR primers | GGTCGTGATTTGATCCTCTG |
| Sequence-based reagent | Ythdc1-R | This paper | PCR primers | TCGAACTCACTCCCATACTC |
| Sequence-based reagent | tubulin-F | This paper | PCR primers | AGCTCAGCACCCTCTGTGTAAT |
| Sequence-based reagent | tubulin-R | This paper | PCR primers | AGCTGGAGCGCATCAATGTGTA |
| Sequence-based reagent | Ime4-F-T3 | This paper | FISH primers | TGTTGGGAAATCACTCCCAATTA AGAGAAGTTTAAGTCCCACGG |
| Sequence-based reagent | Ime4-R-T7 | This paper | FISH primers | GTAATACGACTCACTATAGGGAGA CCACCATCTGGATAACGCTTCTGG |

*Appendix 1 Continued on next page*

*Appendix 1 Continued*

| Reagent type (species) or resource | Designation | Source or reference | Identifiers | Additional information |
|---|---|---|---|---|
| Sequence-based reagent | dMettl14-F-T3 | This paper | FISH primers | TGTTGGGAAATCACTCCCAATTAA CCAATCCGCACAATGACTAC |
| Sequence-based reagent | dMettl14-R-T7 | This paper | FISH primers | GTAATACGACTCACTATAGGGAGA CCACTAGCCAACCTGGTCGAATAC |
| Sequence-based reagent | fl(2)d-F-T3 | This paper | FISH primers | TGTTGGGAAATCACTCCCAATTAA CTAGAGACCATTCACGAGGA |
| Sequence-based reagent | fl(2)d-R-T7 | This paper | FISH primers | GTAATACGACTCACTATAGGGAGACC ACATTATGTATGTCTACGCCGC |
| Sequence-based reagent | vir-F-T3 | This paper | FISH primers | TGTTGGGAAATCACTCCCAATTAA CGAAATGTCGTACAAGGTGC |
| Sequence-based reagent | vir-R-T7 | This paper | FISH primers | GTAATACGACTCACTATAGGGAGACCAC GATCTCCGGGAAAAGTGGTT |
| Sequence-based reagent | nito-F-T3 | This paper | FISH primers | TGTTGGGAAATCACTCCCAATTAA CTCCGATTGATCCCTACGAT |
| Sequence-based reagent | nito-R-T7 | This paper | FISH primers | GTAATACGACTCACTATAGGGAGACC ACGTATCTCCGTCCGTCAAATG |
| Sequence-based reagent | Ythdc1-F-T3 | This paper | FISH primers | TGTTGGGAAATCACTCCCAATTAAC GAATCGAATGGTGGAGACT |
| Sequence-based reagent | Ythdc1-R-T7 | This paper | FISH primers | GTAATACGACTCACTATAGGGAGACCAC CCGTGTGTCTCGGAATAGGT |
| Commercial assay or kit | m6A RNA Methylation Quantification Kit (Colorimetric) | EpiQuik | P-9005–96 | |
| Commercial assay or kit | 2×EasyTaq PCR Super Mix (+Dye) | TransGen Biotech | AS111-12 | |
| Commercial assay or kit | TransTaq-T DNA Polymerase | TransGen Biotech | AP122 | |
| Commercial assay or kit | TransScript one-step gDNA Removal and cDNA Synthesis SuperMix | TransGen Biotech | AT311-03 | |
| Commercial assay or kit | TransStart Tip Green qPCR SuperMix (+Dye II) | TransGen Biotech | AQ142-24 | |
| Chemical compound, drug | Cyanine 3 Tyramide | Akoya Biosciences | FP1046 | |
| Chemical compound, drug | Proteinase K | Sigma | H4784 | |
| Chemical compound, drug | Heparin sodium salt | Sigma | P2308 | |
| Chemical compound, drug | Salmon sperm single-stranded DNA | Sigma | D9156 | |
| Chemical compound, drug | Digoxigenin-11-UTP | Sigma (Roche Diagnostics) | 11209256910 | |
| Chemical compound, drug | Ribonucleoside Triphosphate Set (NTP) | Sigma (Roche Diagnostics) | 11277057001 | |
| Chemical compound, drug | Donkey Serum | Solarbio | N/A | |
| Chemical compound, drug | 5×Transcription buffer | Thermo Scientific | N/A | |

*Appendix 1 Continued*

| Reagent type (species) or resource | Designation | Source or reference | Identifiers | Additional information |
|---|---|---|---|---|
| Chemical compound, drug | T7 RNA polymerase | Thermo Scientific | EP0111 | |
| Software, algorithm | R v4.2.2 | http://www.R-project.org | | |
| Software, algorithm | trim-galore v0.6.10 | https://www.bioinformatics.babraham.ac.uk/projects/trim_galore/ | | |
| Software, algorithm | FastQC v0.12.1 | https://www.bioinformatics.babraham.ac.uk/projects/fastqc/ | | |
| Software, algorithm | HISAT2 v2.2.1 | *Kim et al., 2019* | | |
| Software, algorithm | StringTie v2.1.5 | *Pertea et al., 2015* | | |
| Software, algorithm | exomePeak2 v1.10.0 | *Meng et al., 2013* | | |
| Software, algorithm | HOMER v4.11 | *Heinz et al., 2010* | | |
| Software, algorithm | DiffBind v3.8.4 | *Stark and Brown, 2013* | | |
| Software, algorithm | ChIPseeker v1.34.1 | *Yu et al., 2015* | | |
| Software, algorithm | deepTools v3.5.4 | *Ramírez et al., 2014* | | |
| Software, algorithm | rMATS v4.2.0 | *Shen et al., 2014* | | |
| Software, algorithm | ggplot2 v3.4.0 | *Villanueva and Chen, 2019* | | |
| Software, algorithm | clusterProfiler v4.6.0 | *Wu et al., 2021* | | |
| Software, algorithm | org.Dm.eg.db v3.16.0 | https://www.bioconductor.org/ | | |
| Software, algorithm | DESeq2 v1.38.2 | *Love et al., 2014* | | |
| Software, algorithm | Integrative Genomics Viewer v2.12.0 | *Thorvaldsdóttir et al., 2013* | | |
| Software, algorithm | Fiji v1.53c | *Schindelin et al., 2012* | | |
| Software, algorithm | AnimalTFDB | *Hu et al., 2019* | | |
| Other | DAPI stain | Sigma | 28718-90-3 | (1 µg/mL) |

