## [Editor Report · eLife Assessment]

This **valuable** study suggests that the dosage compensation complex and m6A act in a feedback loop in *Drosophila melanogaster*. The study provides integrated analyses of RNA sequencing and mapping data of the m6A RNA modification in the context of unbalanced genomes, which suggests that m6A modification status may influence H3K16Ac deposition through regulation of the acetyltransferase MOF. However, it is not clear whether this regulation is directly or indirectly related to m6A regulation. The evidence is considered **incomplete** due to technical concerns, as quantitative assessments were made using non-quantitative methods.

---

## [Referee Report · Reviewer #1 (Public review)]

Summary:

This study sought to reveal the potential roles of m6A RNA methylation in gene dosage regulatory mechanisms, particularly in the context of aneuploid genomes in *Drosophila*. Specifically, this work looked at the relationships between expression of m6A regulatory factors, RNA methylation status, classical and inverse dosage effects, and dosage compensation. Using RNA sequencing and m6A mapping experiments, an in depth analysis was performed to reveal changes in m6A status and expression changes across multiple aneuploid *Drosophila* models. The authors propose that m6A methylation regulates MOF and, in turn, deposition of H4K16Ac, critical regulators of gene dosage in the context of genomic imbalance.

Strengths:

This study seeks to address an interesting question with respect to gene dosage regulation and the possible roles of m6A in that process. Previous work has linked m6A to X-inactivation in humans through the Xist lncRNA, and to the regulation of the Sxl in flies. This study seeks to broaden that understanding beyond these specific contexts to more broadly understand how m6A impacts imbalanced genomes in other contexts.

Weaknesses:

The methods being used particularly for analysis of m6A at both the bulk and transcript-specific level are not sufficiently specific or quantitative to be able to confidently draw the conclusions the authors seek to make. MeRIP m6A mapping experiments can be very valuable, but differential methylation is difficult to assess when changes are small (as they often are, in this study but also m6A studies more broadly). For instance based on the data presented and the methods described, it is not clear that the statement that "expression levels at m6A sites in aneuploidies are significantly higher than that in wildtype" is supported. In my initial review I pointed out that MeRIP experiments are not quantitative and can be difficult to interpret when small changes are present. The data as presented still show only RPKM in IP samples, and the text alludes to changes in IP enrichment that are significant but the data do not appear to have been included in the figure. Concerns about the bulk-level m6A measurements also remain, as the new data showing m6A levels in mRNA show changes that are even smaller than those initially demonstrated in total RNA. Yet the data are still presented as significant, biologically relevant changes. The conclusions about mRNA m6A levels are not strengthened by measurements.

---

## [Referee Report · Reviewer #2 (Public review)]

Summary:

The authors have tested effects of partial- or whole-chromosome aneuploidy on the m6A RNA modification in *Drosophila*. The data reveal that overall m6A levels trend up but that the number of sites found by meRIP-seq trend down, which seems to suggest that aneuploidy causes a subset of sites become hyper-methylated. Subsequent bioinformatic analysis of other published datasets establish correlations between activity of the H4K16 acetyltransferase dosage compensation complex (DCC) and expression of m6A components and m6A abundance, suggesting that DCC and m6A can act in a feedback loop. Western blots confirm that Msl2 and MOF alleles alter levels of Mettl3 complex components, but the underlying mechanism remains undefined.

Strengths:

• Thorough bioinformatic analysis of their data

• Incorporation of other published datasets that enhances scope and rigor

• Finds trends that suggest that a chromosome counting mechanism can control m6A, as fits with pub data that the Sxl mRNA is m6A modified in XX females and not XY males

• Provides preliminary evidence that this counting mechanism may be due to DCC effects on expression of m6A components.

Weaknesses:

• The linkage between H4K16 machinery and m6A levels on specific sites remains unclear in this revision.

• The paper relies on m6A comparisons across tissues and developmental stages, which introduces some uncertainty about where and when the DCC-m6A loop acts.

---

## [Author Response]

The following is the authors’ response to the original reviews.

**Public Reviews:**

**Reviewer #1 (Public Review):**
Summary:This study sought to reveal the potential roles of m6A RNA methylation in gene dosage regulatory mechanisms, particularly in the context of aneuploid genomes in *Drosophila*. Specifically, this work looked at the relationships between the expression of m6A regulatory factors, RNA methylation status, classical and inverse dosage effects, and dosage compensation. Using RNA sequencing and m6A mapping experiments, an in-depth analysis was performed to reveal changes in m6A status and expression changes across multiple aneuploid *Drosophila* models. The authors propose that m6A methylation regulates MOF and, in turn, deposition of H4K16Ac, critical regulators of gene dosage in the context of genomic imbalance.Strengths:This study seeks to address an interesting question with respect to gene dosage regulation and the possible roles of m6A in that process. Previous work has linked m6A to X-inactivation in humans through the Xist lncRNA, and to the regulation of the Sxl in flies. This study seeks to broaden that understanding beyond these specific contexts to more broadly understand how m6A impacts imbalanced genomes in other contexts.Weaknesses:The methods being used particularly for analysis of m6A at both the bulk and transcript-specific level are not sufficiently specific or quantitative to be able to confidently draw the conclusions the authors seek to make. MeRIP m6A mapping experiments can be very valuable, but differential methylation is difficult to assess when changes are small (as they often are, in this study but also m6A studies more broadly). For instance, based on the data presented and the methods described, it is not clear that the statement that "expression levels at m6A sites in aneuploidies are significantly higher than that in wildtype" is supported. MeRIP experiments are not quantitative, and since there are far fewer peaks in aneuploidies, it stands to reason that more antibody binding sites may be available to enrich those fewer peaks to a larger extent. But based on the data as presented (figure 2D) this conclusion was drawn from RPKM in IP samples, which may not fully account for changing transcript abundances in absolute (expression level changes) and relative (proportion of transcripts in input RNA sample) terms.

Methylated RNA immunoprecipitation followed by sequencing (MeRIP-seq) is a commonly used strategy of genome-wide mapping of m6A modification. This method uses anti-m6A antibody to immunoprecipitate RNA fragments, which results in selective enrichment of methylated RNA. Then the RNA fragments were subjected to deep sequencing, and the regions enriched in the immunoprecipitate relative to input samples are identified as m6A peaks using the peak calling algorithm. We identified m6A peaks in different samples by the exomePeak2 program and determined common m6A peaks for each genotype based on the intersection of biological replicates. Figure 2D shows the RPM values of m6A peaks in MeRIP samples for each genotype, indicating that the levels of reads in the m6A peak regions were significantly higher in the aneuploid IP samples than in wildtypes. When the enrichment of IP samples relative to Input samples (RPM.IP/RPM.Input) was taken into account, the statistics for all three aneuploidies were still significantly higher than those of the wildtypes (Mann Whitney U test p-values < 0.001). This analysis is not about changes in the abundance of transcripts, but from the MeRIP perspective, showing that there are relatively more m6A-modified reads mapped to the m6A peaks in aneuploidies than that in wildtypes. We hope to provide a possible explanation for the phenomenon that the quantitative changes of m6A peaks are not consistent with the overall m6A abundance trend. We have added the results of IP/Input in the main text, and revised the description in the manuscript to make it more precise to reduce possible misunderstandings.

The bulk-level m6A measurements as performed here also cannot effectively support these conclusions, as they are measured in total RNA. The focus of the work is mRNA m6A regulators, but m6A levels measured from total RNA samples will not reflect mRNA m6A levels as there are other abundance RNAs that contain m6A (including rRNA). As a result, conclusions about mRNA m6A levels from these measurements are not supported.

According to published articles, m6A levels of mRNA or total RNA can be detected by different methods (such as mass spectrometry, 2D thin-layer chromatography, etc.) in *Drosophila* cells or tissues [1-3]. We used the EpiQuik m6A RNA Methylation Quantification Kit, which is suitable for detecting m6A methylation status directly using total RNA isolated from any species such as mammals, plants, fungi, bacteria, and viruses. This kit has previously been used by researchers to detect the m6A/A ratio in total RNA [4, 5] or purified mRNA [6] from different species. Our pre-experiments showed that the enrichment of mRNA from total RNA did not appear to significantly affect the results of the detection of m6A levels.

We extracted and purified mRNA from the heads of the control and MSL2 transgenic *Drosophila* to verify our conclusion. mRNA was isolated from total RNA using the Dynabeads mRNA purification kit (Invitrogen, Carlsbad, CA, USA, 61006). It was showing a heightened abundance of m6A modification on mRNA as opposed to total RNA (Figure 7E,F; Figure 7—figure supplement 1G,H). Compared with control *Drosophila*, the abundance changes of m6A in mRNA and total RNA in MSL2 transgenic *Drosophila* are basically the same. These results supported the conclusions in our manuscript. In the MSL2 knockdown *Drosophila*, the m6A modification levels on mRNA mirrored those observed on total RNA, exhibiting a significant downregulation (Figure 7E; Figure 7—figure supplement 1G). The only difference is that no substantial difference in the m6A abundance on mRNA was detected between MSL2 overexpressed female and the control *Drosophila* (Figure 7F; Figure 7—figure supplement 1H). It is suggested that m6A modification in other types of RNA other than mRNA (e.g., lncRNA, rRNA) is not necessarily meaningless, which is the future research direction. We will also add discussions of this issue in the manuscript.

(1) Lence T, et al. (2016) m6A modulates neuronal functions and sex determination in *Drosophila*. Nature 540(7632):242-247.

(2) Haussmann IU, et al. (2016) m(6)A potentiates Sxl alternative pre-mRNA splicing for robust *Drosophila* sex determination. Nature 540(7632):301-304.

(3) Kan L, et al. (2017) The m(6)A pathway facilitates sex determination in *Drosophila*. Nat Commun 8:15737.

(4) Zhu C, et al. (2023) RNA Methylome Reveals the m(6)A-mediated Regulation of Flavor Metabolites in Tea Leaves under Solar-withering. Genomics Proteomics Bioinformatics 21(4):769-787.

(5) Song H, et al. (2021) METTL3-mediated m(6)A RNA methylation promotes the anti-tumour immunity of natural killer cells. Nat Commun 12(1):5522.

(6) Yin H, et al. (2021) RNA m6A methylation orchestrates cancer growth and metastasis via macrophage reprogramming. Nat Commun 12(1):1394.

**Reviewer #2 (Public Review):**
Summary:The authors have tested the effects of partial- or whole-chromosome aneuploidy on the m6A RNA modification in *Drosophila*. The data reveal that overall m6A levels trend up but that the number of sites found by meRIP-seq trend down, which seems to suggest that aneuploidy causes a subset of sites to become hyper-methylated. Subsequent bioinformatic analysis of other published datasets establish correlations between the activity of the H4K16 acetyltransferase dosage compensation complex (DCC) and the expression of m6A components and m6A abundance, suggesting that DCC and m6A can act in a feedback loop on each other. Overall, this paper uses bioinformatic trends to generate a candidate model of feedback between DCC and m6A. It would be improved by functional studies that validate the effect in vivo.Strengths:• Thorough bioinformatic analysis of their data.• Incorporation of other published datasets that enhance scope and rigor.• Finds trends that suggest that a chromosome counting mechanism can control m6A, as fits with pub data that the Sxl mRNA is m6A modified in XX females and not XY males.• Suggests this counting mechanism may be due to the effect of chromatin-dependent effects on the expression of m6A components.Weaknesses:• The linkage between H4K16 machinery and m6A is indirect and based on bioinformatic trends with little follow-up to test the mechanistic bases of these trends.

Western blots were performed to detect H4K16Ac in Ythdc1 knockdown *Drosophila* and control *Drosophila*. Through quantitative analysis, it is demonstrated that H4K16Ac levels changed significantly in Ythdc1 knockdown *Drosophila*. Combined with the results of polytene chromosome immunostaining in third instar larvae, we found that Ythdc1 affects the expression of H4K16Ac in tissue- and developmental stage-specific manners. This specificity may be associated with the onuniformity and heterogeneity of RNA m6A modification characteristics, encompassing the tissue specificity, the developmental specificity, the different numbers of m6A sites in one transcript, the different proportions of methylated transcripts, et cetera [1-3].

In addition, we found a set of ChIP-seq data (GSE109901) of H4K16ac in female and male *Drosophila* larvae from the public database, and analyzed whether H4K16ac is directly associated with m6A regulator genes. ChIP-seq is a standard method to study transcription factor binding and histone modification by using efficient and specific antibodies for immunoprecipitation. The results showed that there were H4K16ac peaks at the 5' region in gene of m6A reader Ythdc1 in both males and females. In addition, most of the genome sites where the other m6A regulator genes located are acetylated at H4K16 in both sexes, except that Ime4 shows sexual dimorphism and only contains H4K16ac peak in females. These results indicate that the m6A regulator gene itself is acetylated at H4K16, so there is a direct relationship between H4K16ac and m6A regulators. We have added these contents to the text.

Our analysis of experimental outcomes and public sequencing data has shed light on the interaction of the m6A reader protein Ythdc1 with H4K16Ac. We appreciate your interest in the complex interplay between H4K16Ac and m6A modifications. We acknowledge the intricacy of this interaction and concur that it merits further investigation, potentially supported by additional experiments.

In current submitted manuscript, it is mainly focused on the role of RNA m6A modification in genomes experiencing imbalance, and we are going to explore this complex interplay in subsequent work for sure.

(1) Meyer, K. D., et al. (2012). Comprehensive analysis of mRNA methylation reveals enrichment in 3' UTRs and near stop codons. Cell, 149(7), 1635-1646.

(2) Meyer, K. D., & Jaffrey, S. R. (2014). The dynamic epitranscriptome: N6-methyladenosine and gene expression control. Nature Reviews: Molecular Cell Biology, 15(5), 313-326.

(3) Zaccara, S., Ries, R. J., & Jaffrey, S. R. (2019). Reading, writing and erasing mRNA methylation. Nature Reviews: Molecular Cell Biology, 20(10), 608-624.

• The paper lacks sufficient in vivo validation of the effects of DCC alleles on m6A and vice versa. For example, Is the Ythdc1 genomic locus a direct target of the DCC component Msl-2 ? (see Figure 7).

In order to study whether Ythdc1 genomic locus is a direct target of DCC component, we first analyzed a published MSL2 ChIP-seq data of *Drosophila* (GSE58768). Since MSL2 is only expressed in males under normal conditions, this set of data is from male *Drosophila*. According to the results, the majority (99.1%) of MSL2 peaks are located on the X chromosome, while the MSL2 peaks on other chromosomes are few. This is consistent with the fact that MSL2 is enriched on the X chromosome in male *Drosophila* [1, 2]. Ythdc1 gene is located on chromosome 3L, and there is no MSL2 peak near it. Similarly, other m6A regulator genes are not X-linked, and there is no MSL2 peak. Then we analyzed the MOF ChIP-seq data (GSE58768) of male *Drosophila*. It was found that 61.6% of MOF peaks were located on the X chromosome, which was also expected [3, 4]. Although there are more MOF peaks on autosomes than MSL2 peaks, MOF peaks are absent on m6A regulator genes on autosomes. Therefore, at present, there is no evidence that the gene locus of m6A regulators are the direct targets of DCC component MSL2 and MOF, which may be due to the fact that most MSL2 and MOF are tethered to the X chromosome by MSL complex under physiological conditions. Whether there are other direct or indirect interactions between Ythdc1 and MSL2 is an issue worthy of further study in the future.

(1) Bashaw GJ & Baker BS (1995) The msl-2 dosage compensation gene of *Drosophila* encodes a putative DNA-binding protein whose expression is sex specifically regulated by Sex-lethal. Development 121(10):3245-3258.

(2) Kelley RL, et al. (1995) Expression of msl-2 causes assembly of dosage compensation regulators on the X chromosomes and female lethality in *Drosophila*. Cell 81(6):867-877.

(3) Kind J, et al. (2008) Genome-wide analysis reveals MOF as a key regulator of dosage compensation and gene expression in *Drosophila*. Cell 133(5):813-828.

(4) Conrad T, et al. (2012) The MOF chromobarrel domain controls genome-wide H4K16 acetylation and spreading of the MSL complex. Dev Cell 22(3):610-624.

Quite a bit of technical detail is omitted from the main text, making it difficult for the reader to interpret outcomes.

(1) Please add the tissues to the labels in Figure 1D.

Figure 1D shows the subcellular localization of FISH probe signals in *Drosophila* embryos. Arrowheads indicate the foci of probe signals. The corresponding tissue types are (1) blastoderm nuclei; (2) yolk plasm and pole cells; (3) brain and midgut; (4) salivary gland and midgut; (5) blastoderm nuclei and yolk cortex; (6) blastoderm nuclei and pole cells; (7) blastoderm nuclei and yolk cortex; (8) germ band. We have added these to the manuscript.

(2) In the main text, please provide detail on the source tissues used for meRIP; was it whole larvae? adult heads? Most published datasets are from S2 cells or adult heads and comparing m6A across tissues and developmental stages could introduce quite a bit of variability, even in wt samples. This issue seems to be what the authors discuss in lines 197-199.

In this article, the material used to perform MeRIP-seq was the whole third instar larvae. Because trisomy 2L and metafemale *Drosophila* died before developing into adults, it was not possible to use the heads of adults for MeRIP-seq detection of aneuploidy. For other experiments described here, the m6A abundance was measured using whole larvae or adult heads; material used for RT-qPCR analysis was whole larvae, larval brains, or adult heads; *Drosophila* embryos at different developmental stages were used for fluorescence in situ hybridization (FISH) experiments. We provide a detailed description of the experimental material for each assay in the manuscript.

(3) In the main text, please identify the technique used to measure "total m6A/A" in Fig 2A. I assume it is mass spec.

We used the EpiQuik m6A RNA Methylation Quantification Kit (Colorimetric) (Epigentek, NY, USA, Cat # P-9005) to measure the m6A/A ratio in RNA samples. This kit is commercially available for quantification of m6A RNA methylation, which used colorimetric assay with easy-to-follow steps for convenience and speed, and is suitable for detecting m6A methylation status directly using total RNA isolated from any species such as mammals, plants, fungi, bacteria, and viruses.

(4) Line 190-191: the text describes annotating m6A sites by "nearest gene" which is confusing. The sites are mapped in RNAs, so the authors must unambiguously know the identity of the gene/transcript, right?

When the m6A peaks were annotated using the R package ChIPseeker, it will include two items: "genomic annotation" and "nearest gene annotation". "Genomic annotation" tells us which genomic features the peak is annotated to, such as 5’UTR, 3’UTR, exon, etc. "Nearest gene annotation" indicates which specific gene/transcript the peak is matched to. We modified the description in the main text to make it easier to understand.

**Recommendations for the authors:**

**Reviewer #1 (Recommendations For The Authors):**
While I believe this study aims to address a very interesting question and demonstrates intriguing evidence suggesting a role for m6A in unbalanced genomes, technical limitations in the methods being used limited my confidence in the overall conclusions. In addition, some of the analyses seemed to distract a bit from the main question of the work, which made thoroughly reading and reviewing the work challenging at times due to the length and lack of cohesion. Some specific points and suggestions are detailed below.(1) Some specific points/recommendations for the bulk m6A measurements: for Figure 2A, the authors refer to m6A/A ratio in the text, but based on the methods section and axis labels in Figure 2A (as well as other figures), it may represent m6A% in total RNA. The authors should just clarify which one it is and make the text and figures consistent. The methods description also seems to specify that m6A is quantified in total RNA, and yet the factors being discussed (Ime4, Ythdc1, etc) are associated with m6A in mRNA. Since m6A is present in non-mRNAs (including highly abundant rRNAs), m6A analysis of total RNA may be masking some of the effects due to the relatively low abundance of mRNA relative to rRNA. It is possible that the above point contributes to the discrepancy between the overall m6A abundance in aneuploidies and the changing methylase expression levels (which does seem to correlate better with m6A sequencing data). On a related note, though the authors suggest in Figures 7E and F that m6A level changes are different in males and females, the levels and trends of m6A% in these panels seem quite similar, and the absence of the presence of statistical significance seems driven by higher variation (larger error bars) in the measurements in 7F (and again effects may be masked if total RNA is being quantified). This may be a very addressable issue, as m6A analysis of mRNA-enriched samples should be feasible, and in fact, may show clearer changes to better support the authors' conclusions.

Thank you for your helpful comments.

As suggested, the abundance of m6A on mRNA were detected (Figure 7E, F). Total RNA was extracted from the heads of the control and MSL2 transgenic *Drosophila* and mRNA was isolated using the Dynabeads mRNA purification kit (Invitrogen, Carlsbad, CA, USA, 61006). 300-600 ng mRNA can be purified from 40 μg total RNA (200-300 heads per sample). We used the EpiQuik m6A RNA Methylation Quantification Kit (Colorimetric) (Epigentek, NY, USA, Cat # P-9005) to measure the abundance of m6A in mRNA samples (200ng). The results obtained by this method represent the m6A/A ratio (%), which is also written as m6A% on the user guide of the kit. We made corresponding revisions in the main text and figures to made them consistent.

It is showing a heightened abundance of m6A modification on mRNA as opposed to total RNA including some other types of RNA such as mRNA, lncRNA, and rRNA (Figure 7E,F; Figure 7—figure supplement 1G,H). Consistently, in the MSL2 knockdown *Drosophila*, the m6A modification levels on mRNA mirrored those observed on total RNA, exhibiting a significant downregulation (Figure 7E; Figure 7—figure supplement 1G). In contrast, no substantial difference in the m6A abundance on mRNA was detected between MSL2 overexpressed *Drosophila* and the control *Drosophila* (Figure 7F; Figure 7—figure supplement 1H). The differences of m6A abundance between males and females were not statistically significant (Figure 7E,F), prompting us to make revisions to the manuscript.

(2) The analyses in Figures 5 and 6 describe a lot of different comparisons derived from these datasets, and while there seem to be many interesting new hypotheses to be tested, the authors do not make any definitive conclusions from these analyses. These figures also seem to diverge a bit from the main conclusion of the work, and from this reviewer's perspective made it more difficult to read and review the work. Overall streamlining the narrative may help readers appreciate the main conclusions of the work (though this is of course up to the author's discretion).

As indicated in Figure 5, the results demonstrated a sexually dimorphic role of m6A modification in the regulation of gene expression in aneuploid *Drosophila*, suggesting its potential involvement in the gene regulatory network through interactions with dosage-sensitive regulators. Furthermore, Figure 6 illustrated the intricate interplay between RNA m6A modification, gene expression, and alternative splicing under genomic imbalance, with RNA splicing being more intimately associated with m6A methylation than gene transcription itself.

This manuscript also discussed the correlation between methylation status and classical dosage effects, dosage compensation effects, and inverse dosage effects. We have initially demonstrated that RNA m6A methylation could influence dosage-dependent gene regulation via multiple avenues, such as interactions with dosage-sensitive modifiers, alternative splicing mechanisms, the MSL complex, and other related processes. Indeed, our study primarily utilizes m6A methylated RNA immunoprecipitation sequencing (MeRIP-Seq) to comprehensively investigate the role of RNA m6A modification in genomes experiencing imbalance. We agree that more specific and in-depth research on these factors will be instrumental in elucidating the precise mechanisms by which m6A modification regulates expression in unbalanced genomes, which we acknowledge as a significant avenue for our future research.

We are grateful for your suggestions and, should it be necessary, we might to simplify the volume of the whole manuscript by removing or condensing the data analyse and description to enhance the prominence of the central theme.

**Reviewer #2 (Recommendations For The Authors):**
Overall, please provide enough technical detail in the main text so that the reader understands what was done, and does not have to repeatedly dig into figure legends and materials and methods to understand each data statement.

Thank you for your suggestions. We have added some technical details to the manuscript and made some modifications as suggested.